# Identification of Analytic Nonlinear Dynamical Systems with Non-asymptotic Guarantees

**Negin Musavi**
nmusavi2@illinois.edu

**Ziyao Guo**
ziyaog2@illinois.edu

**Geir Dullerud**
dullerud@illinois.edu

**Yingying Li**
yl101@illinois.edu

Coordinated Science Laboratory
University of Illinois Urbana-Champaign

## Abstract

This paper focuses on the system identification of an important class of nonlinear systems: nonlinear systems that are linearly parameterized, which enjoy wide applications in robotics and other mechanical systems. We consider two system identification methods: least-squares estimation (LSE), which is a point estimation method; and set-membership estimation (SME), which estimates an uncertainty set that contains the true parameters. We provide non-asymptotic convergence rates for LSE and SME under i.i.d. control inputs and control policies with i.i.d. random perturbations, both of which are considered as non-active-exploration inputs. Compared with the counter-example based on piecewise-affine systems in the literature, the success of non-active exploration in our setting relies on a key assumption about the system dynamics: we require the system functions to be real-analytic. Our results, together with the piecewise-affine counter-example, reveal the importance of differentiability in nonlinear system identification through non-active exploration. Lastly, we numerically compare our theoretical bounds with the empirical performance of LSE and SME on a pendulum example and a quadrotor example.

## 1  Introduction

Learning control-dynamical systems with statistical methodology has received significant attention in the past decade (Sarker et al., 2023; Li et al., 2023b; Chen and Hazan, 2021; Simchowitz and Foster, 2020; Wagenmaker and Jamieson, 2020; Simchowitz et al., 2018; Dean et al., 2018; Abbasi-Yadkori and Szepesvári, 2011; Li et al., 2021b). In particular, the estimation of linear dynamical systems, e.g. $x_{t+1} = A^* x_t + B^* u_t + w_t$, is relatively well-studied: it has been shown that non-active exploration by i.i.d. noises on control inputs $u_t$ and system disturbances $w_t$ are already enough for accurate system identification, and least square estimation (LSE) can achieve the optimal estimation convergence rate (Simchowitz and Foster, 2020; Simchowitz et al., 2018).

However, nonlinear control systems are ubiquitous in real-world applications, e.g. robotics (Siciliano et al., 2010; Alaimo et al., 2013), power systems (Simpson-Porco et al., 2016), transportation (Kong et al., 2015), etc. Motivated by this, there has been a lot of attention on learning nonlinear systems recently. One natural and popular direction to study nonlinear system identification is on learning linearly parameterized nonlinear systems as defined below, which is a straightforward extension from

38th Conference on Neural Information Processing Systems (NeurIPS 2024).

the standard linear systems (Mania et al., 2022; Khosravi, 2023; Foster et al., 2020)

$$x_{t+1} = \theta_* \phi(x_t, u_t) + w_t$$

where $\theta_*$ is a vector of unknown parameters and $\phi(x_t, u_t)$ is a known vector of nonlinear features.

On the one hand, some classes of these systems are shown to enjoy similar benefits of linear systems. For example, bilinear systems can also be estimated by LSE under non-active exploration with i.i.d. noises (Sattar et al., 2022), as well as linear systems with randomly perturbed nonlinear policies (Li et al., 2023b).

On the other hand, it is also known that non-active exploration is insufficient for general linearly parameterized nonlinear systems. In particular, (Mania et al., 2022) provides a counter example showing that non-active exploration is insufficient to learn accurate models under piece-wise affine feature functions. This motivates a sequence of follow-up work on the design of active exploration for nonlinear system estimation, which is largely motivated by the non-smooth feature functions such as ReLu in neural networks (Mania et al., 2022; Kowshik et al., 2021; Khosravi, 2023).

However, there is a big gap between bilinear systems, which is infinitely differentiable, and the counter example by non-smooth systems. A natural question is: to what extent can non-active exploration still work for linearly parameterized nonlinear systems?

**Contributions.** One major contribution of this paper is showing that LSE with non-active i.i.d. noises can efficiently learn any linearly parameterized nonlinear systems with real-analytic feature functions and provide a non-asymptotic convergence rate. Notice that real-analytic feature functions are common in physical systems. For example, polynomial systems satisfy this requirement and have wide applications in power systems (Simpson-Porco et al., 2016), fluid dynamics (Noack et al., 2003), etc. Further, trigonometric functions also satisfy the real-analytic property so a large range of robotics and mechanical systems also satisfy this requirement (Siciliano et al., 2010; Alaimo et al., 2013).

A side product of our LSE convergence rate analysis is the convergence rate for another commonly used uncertainty quantification method in control: set membership estimation (SME).

Numerically, we test our theoretical results in pendulum and quadrotor systems. Simulations show that LSE and SME can indeed efficiently explore the system and converge to the true parameter under non-active exploration noises.

Technically, the key step in our proof is establishing the block-martingale-small-ball condition (BMSB) for general analytic feature functions, which greatly generalizes the bilinear feature function in Sattar et al. (2022). Our result is built on an intuition inspired by the counter example in (Mania et al., 2022): the counter example in (Mania et al., 2022) requires that some feature function is zero in a certain region, so nothing can be learned about its parameter if the states stay in this region. However, analytic functions cannot be a constant zero in a positive-measure region unless it is a constant zero everywhere. Therefore, the counter example does not work, and non-active exploration around any states can provide some useful information. Our proof formalizes this intuition by utilizing the Paley-Zygmund Petrov inequality (Petrov, 2007).

**Related work.** Inspired by neural network parameterization, nonlinear systems of the form $x_{t+1} = \phi(A_* x_t) + w_t$ is also studied in the literature, where $\phi(\cdot)$ is a known nonlinear link function and $A_*$ is unknown. The least square cost is no longer quadratic or even convex in this case and various optimization methods have been proposed to learn this type of systems (Kowshik et al., 2021; Sattar et al., 2022; Foster et al., 2020).

Another related line of research focuses on nonlinear regression with dependent data (Ziemann and Tu, 2022; Ziemann et al., 2023, 2024),[1] which can be applied to nonlinear system identification. The nonlinear regression in (Ziemann and Tu, 2022; Ziemann et al., 2023, 2024) is based on non-parametric LSE and its variants, and their convergence rates under different scenarios have been analyzed. It is interesting to note that this line of work usually assumes certain persistent excitation assumptions,[2] whereas our paper demonstrates that persistent excitation holds by establishing the BMSB condition for linearly parameterized and real-analytic nonlinear control systems.

---

[1] $y_t = f_*(x_t) + w_t$ is considered, where $x_t$ and $y_t$ correlate with the historical data.

[2] For example, (Ziemann and Tu, 2022) assumes hyper-contractivity, and (Ziemann et al., 2024) assumes the empirical covariance of the $\{x_t\}_{t \geq 0}$ process is invertible with high probability (Corollary 3.2).

Uncertainty set estimation is crucial for robust control under model uncertainties Lu and Cannon (2023); Lorenzen et al. (2019); Li et al. (2021a). SME is a widely adopted uncertainty set estimation method in robust adaptive control (Lorenzen et al., 2019; Lu and Cannon, 2023; Bertsekas, 1971; Bai et al., 1995). Recently, there is an emerging interest in analyzing SME's convergence and convergence rate for dynamical systems (Li et al., 2024; Lu et al., 2019; Xu and Li, 2024), because previous analysis focus more on the linear regression problem (e.g. (Akçay, 2004; Bai et al., 1998)). There are also recent applications of SME to online control Yu et al. (2023), power systems Yeh et al. (2024), and computer vision Gao et al. (2024); Tang et al. (2024).

**Notation.** The set of non-negative real numbers is denoted by $\mathbb{R}_{\geq 0}$. The notation $\lceil \cdot \rceil$ stands for the ceiling function. For a real vector $z \in \mathbb{R}^n$, $\|z\|_2$ represents its $\ell_2$ norm, $\|z\|_\infty$ represents its $\ell_\infty$ norm, and $z^i$ represents its $i$-th component with $i = 1 \cdots n$. The set of real symmetric matrices is denoted by $\mathbb{S}^n$. For a real matrix $Z$, $Z^\mathsf{T}$ represents its transpose, $\|Z\|_2$ its maximum singular value, $\|Z\|_F$ its Frobenius norm, $\sigma_{\min}(Z)$ its minimum singular value, $\mathrm{vec}(Z)$ its vectorization obtained by stacking its columns, and for a real square matrix $Z$, $\mathrm{tr}(Z)$ represents its trace. For a real symmetric matrix $Z$, $Z \succ 0$ and $Z \succeq 0$ indicate that $Z$ is positive definite and positive semi-definite, respectively. For a measurable set $\mathcal{E} \subset \mathbb{R}^n$, $\lambda^n(\mathcal{E})$ represents its Lebesgue measure in $\mathbb{R}^n$ and $\mathcal{E}^c$ represents its complement in $\mathbb{R}^n$. The notation $\emptyset$ stands for the empty set. For a set $\mathcal{T}$ of matrices $\theta \in \mathbb{R}^{n \times m}$, $\mathrm{diam}(\mathcal{T})$ denotes its diameter and it is defined as $\mathrm{diam}(\mathcal{T}) = \sup_{\theta, \theta' \in \mathcal{T}} \|\theta - \theta'\|_F$. For $z_i \in \mathbb{R}$ with $i = 1, \cdots, \ell$, the notation $\mathrm{diag}(z_1, \cdots, z_\ell)$ denotes a matrix in $\mathbb{R}^{\ell \times \ell}$ with diagonal entries of $z_i$. This paper uses `truncated-Gaussian`$(0, \sigma_w, [-w_{\max}, w_{\max}])$ to refer to the truncated Gaussian distribution generated by Gaussian distribution with zero mean and $\sigma_w^2$ variance with truncated range $[-w_{\max}, w_{\max}]$. The same applies to multi-variate truncated Gaussian distributions.

## 2 Problem Formulation and Preliminaries

This paper studies the system identification/estimation of linearly parameterized nonlinear systems:

$$x_{t+1} = \theta_* \phi(x_t, u_t) + w_t, \tag{1}$$

where $x_t \in \mathbb{R}^{n_x}$, $u_t \in \mathbb{R}^{n_u}$, and $w_t \in \mathbb{R}^{n_x}$ denote the state, control input, and system disturbance respectively; $\theta_* \in \mathbb{R}^{n_x \times n_\phi}$ denotes the unknown parameters to be estimated, and $\phi(\cdot)$ denotes a vector of known nonlinear feature/basis functions, i.e., $\phi(\cdot) = (\phi^1(\cdot), \cdots, \phi^{n_\phi}(\cdot))^\mathsf{T}$, where $\phi^i(\cdot) : \mathbb{R}^{n_x + n_u} \to \mathbb{R}$. Without loss of generality, we consider zero initial condition, i.e. $x_0 = 0$, and linearly independent feature functions, that is, $\sum_{i=1}^{n_\phi} c_i \phi^i(x_t, u_t) = 0$ implies that $c_i = 0$ for all $i$.[3]

The linearly parameterized nonlinear system (1) is a natural generalization of linear control systems $x_{t+1} = A_* x_t + B_* u_t + w_t$ and has wide applications in, for example, robotics (Siciliano et al., 2010; Alaimo et al., 2013), power systems (Simpson-Porco et al., 2016), transportation (Kong et al., 2015), etc. Therefore, there has been a lot of research on learning this type of system (1) utilizing the methodology and insights from linear system estimation. For example, it is common to estimate a linearly parameterized nonlinear system by least squares estimation (LSE), which enjoys desirable performance in linear systems.

In particular, LSE for (1) is reviewed below

$$\hat{\theta}_T = \arg\min_{\hat{\theta}} \sum_{t=0}^{T-1} \left\| x_{t+1} - \hat{\theta} \phi(x_t, u_t) \right\|_2^2. \tag{2}$$

For linear systems, LSE enjoys the following good property: LSE can achieve the optimal rate of convergence with i.i.d. noises $w_t$ and i.i.d. control inputs $u_t$ under proper conditions ( (Simchowitz et al., 2018)). This good property has been generalized to some linearly parameterized nonlinear systems, such as bilinear systems, and linear systems with nonlinear control policies. Unfortunately, general linearly parameterized nonlinear systems do not enjoy this good property of linear systems, meaning i.i.d. random inputs may not provide enough exploration for non-smooth feature functions $\phi(\cdot)$. Therefore, a series of follow-up work focuses on the design of active exploration methods.

However, due to the simplicity of implementation, i.i.d. random inputs remain a popular method in empirical research of system identification and sometimes enjoy satisfactory performance, despite

---

[3]If the features are not independent, they can be converted to independent ones since the features are known.

the lack of theoretical guarantees. Therefore, this paper aims to establish more general conditions that allow provable convergence of nonlinear system estimation under i.i.d. random inputs.

In the rest of this paper, we will show that with certain smoothness and continuous conditions, i.i.d. random inputs are sufficient for estimation of (1), which recovers the good property of linear systems.

## 2.1 Assumptions

In the following, we formally describe the smoothness and continuity conditions that enables efficient exploration of (1) by i.i.d. random inputs.

**Assumption 1** (Analytic feature functions). *All components of the feature vector $\phi(\cdot)$ are real analytic functions in $\mathbb{R}^{n_x+n_u}$,[4] i.e., for every $1 \leq i \leq n_\phi$, $\phi^i(x, u)$ is an infinitely differentiable function such that the Taylor expansion in every $(\bar{x}, \bar{u})$ converges point-wise to $\phi^i(x, u)$ in a neighborhood of $(\bar{x}, \bar{u})$.*

Analytic functions include polynomial functions and trigonometric functions, which are important components of many physical systems in real-world applications, e.g. power systems, robotics, transportation systems, etc. In particular, we provide two illustrative examples below.

**Example 1** (Pendulum). *Many multilink robotic manipulators can be understood as interconnected pendulum dynamics. The motion equations of a single pendulum, consisting of a mass $m$ suspended from a weightless rod of length $l$ fixed at a pivot without friction, can be expressed as:*

$$\ddot{\alpha} = -\frac{g}{l}\sin(\alpha) + \frac{u}{ml^2} + w,$$

*where $\alpha$ represents the angle of the rod relative to the vertical axis, $g$ is the gravity constant, $u$ is the torque input, and $w$ is the disturbance applied to this system. After discretization the system dynamics can be rewritten in the structure of (1) with the feature vector consisting of expressions involving $\sin(\alpha)$ and $u$, all of which are analytic functions. The matrix of unknown parameters contains terms of the pendulum's mass and the rod's length.*

**Example 2** (Quadrotor (Alaimo et al., 2013)). *Let $p \in \mathbb{R}^3$ and $v \in \mathbb{R}^3$ represent the center of mass position and velocity of the quadrotor in the inertial frame, respectively; let $\omega \in \mathbb{R}^3$ denote its angular velocity in the body-fixed frame and $q \in \mathbb{R}^4$ denote the quaternion vector. The quadrotor's equations of motion can then be expressed as:*

$$\frac{d}{dt}\begin{pmatrix} p \\ v \\ q \\ \omega \end{pmatrix} = \begin{pmatrix} v \\ -ge_z + \frac{1}{m}Qf_u \\ \frac{1}{2}\Omega q \\ I^{-1}(\tau_u - \omega \times I\omega) \end{pmatrix} + w,$$

*where $g$ is the gravity constant, $m$ is its total mass, $I = \mathrm{diag}(I_{xx}, I_{yy}, I_{zz})$ its inertia matrix with respect to the body-fixed frame, $f_u \in \mathbb{R}$ the total thrust, $\tau_u \in \mathbb{R}^3$ the total moment in the body-fixed frame, $e_z = (0, 0, 1)^\intercal$, $Q = \begin{pmatrix} q_0^2 + q_1^2 - q_2^2 - q_3^2 & 2(q_1q_2 - q_0q_3) & 2(q_0q_2 - q_1q_3) \\ 2(q_1q_2 - q_0q_3) & q_0^2 - q_1^2 + q_2^2 - q_3^2 & 2(q_2q_3 - q_0q_1) \\ 2(q_1q_3 - q_0q_2) & 2(q_0q_1 - q_2q_3) & q_0^2 - q_1^2 - q_2^2 + q_3^2 \end{pmatrix}$, and $\Omega = \begin{pmatrix} 0 & -\omega_1 & -\omega_2 & -\omega_3 \\ \omega_1 & 0 & \omega_3 & -\omega_2 \\ \omega_2 & -\omega_3 & 0 & \omega_1 \\ \omega_3 & \omega_2 & -\omega_1 & 0 \end{pmatrix}.$*

*Similar to the pendulum example, after discretization the system dynamics can be rewritten in the structure of (1) with the feature vector consisting of* cubic polynomials *in states and inputs, which are real-analytic. The unknown parameters contain terms of the mass and inertial moments of the quadrotor.*

Next, we introduce the assumption on the random inputs, which relies on the following definition.

**Definition 1** (Semi-continuous distribution). *We define a probability distribution $\mathbb{P}$ as semi-continuous if there does not exist a set $E$ with Lebesgue measure zero such that $\mathbb{P}(E) = 1$.*

---

[4]This assumption can be relaxed to locally analytic functions in a large enough bounded set.

The semi-continuous distribution is a weaker requirement than continuous distributions. In particular, any continuous distributions, or a mixture distribution with one component as a continuous distribution, can satisfy the requirement of semi-continuity. The semi-continuity can also be interpreted by the Lebesgue Decomposition Theorem (Chapter 6 of (Halmos, 2013)) as discussed below.

**Remark 1** (Connection with Lebesgue Decomposition Theorem). *Definition 1 can be interpreted by the Lebesgue Decomposition Theorem, which suggests that any probability distribution can be decomposed into a purely atomic component and a non-atomic component (see more details in Halmos (2013)). A semi-continuous distribution as defined in Definition 1 requires the distribution's non-atomic component to be nonzero.*

In the following, we provide the assumptions on $w_t$ and $u_t$ using the semi-continuity definition.

**Assumption 2** (Bounded i.i.d. and semi-continuous disturbance). *$w_t$ is i.i.d. following a semi-continuous distribution with zero mean and a positive definite covariance matrix $\Sigma_w \succeq \sigma_w^2 I_{n_x} \succ 0$ and a bounded support, i.e. $\|w_t\|_\infty \leq w_{\max}$ almost surely for all $t$.*

The i.i.d. assumption is common in the literature of system identification for linear and nonlinear systems. As for the bounded assumption on $w_t$, although it is stronger than the sub-Gaussian assumption on $w_t$ in the literature of linear system estimation, it is a common assumption in the literature of nonlinear system estimation (Mania et al., 2022; Shi et al., 2021; Kim and Lavaei, 2024). Further, in many physical applications, noises are usually bounded, e.g. the wind disturbances in quadrotor systems are bounded, the renewable energy injections in power systems are also bounded, etc.

The semi-continuity assumption may seem restrictive since it rules out the discrete distributions. However, the disturbances in many realistic systems can satisfy the semi-continuity because realistic noises are usually generated from a mixture distribution where at least one component is continuous, e.g. the wind disturbances and renewable generations are continuous.

For the control inputs $u_t$, we first impose the same assumption as Assumption 2 for simplicity. Later in Section 3, we will also discuss the relaxation of this assumption to include control policies.

**Assumption 3** (Bounded i.i.d. and semi-continuous inputs). *$u_t$ is i.i.d. following a semi-continuous distribution with zero mean and positive definite covariance $\Sigma_u \succeq \sigma_u^2 I_{n_x} \succ 0$ and bounded support, i.e. $\|u_t\|_\infty \leq u_{\max}$ almost surely for all $t$.*

Lastly, we introduce our stability assumption based on the input-to-state stability definition below.

**Definition 2** (Locally input-to-state stability (LISS)). *Consider the general nonlinear system $x_{t+1} = f(x_t, d_t)$ with $x_t \in \mathbb{R}^{n_x}$, $d_t \in \mathbb{R}^{n_d}$, $f$ being a continuous function such that $f(0,0) = 0$. This system is called locally input-to-state stable (LISS) if there exist constants $\rho_x > 0$, $\rho > 0$ and functions $\gamma \in \mathcal{K}$, $\beta \in \mathcal{KL}$ such that for all $x_0 \in \{x_0 \in \mathbb{R}^{n_x} : \|x_0\|_2 \leq \rho_x\}$ and any input $d_t \in \{d \in \mathbb{R}^{n_d} : \sup_t \|d_t\|_\infty \leq \rho\}$, it holds that $\|x_t\|_2 \leq \beta\big(\|x_0\|_2, t\big) + \gamma\big(\sup_t \|d_t\|_\infty\big)$ for all $t \geq 0$.*[5]

**Assumption 4** (LISS system). *System (1) is LISS with parameters $\rho_x$ and $\rho$ such that $\rho_x \geq \|x_0\|_2$ and $\rho \geq \max(w_{\max}, u_{\max})$, respectively.*

Assumption 4 is imposed, together with the bounded disturbances and inputs in Assumptions 2 and 3, to guarantee bounded states during the control dynamics (for instance, see the proof of Theorem 1 in Appendix A). In particular, many studies on learning-based nonlinear control require a certain boundedness on the states for theoretical analysis Sattar and Oymak (2022); Foster et al. (2020); Li et al. (2023a).

In addition, it is interesting to note that this paper only requires local stability of the dynamics, whereas several learning-based nonlinear control papers assume certain global properties, such as global exponential stability in (Foster et al., 2020), global exponential incremental stability in (Sattar and Oymak, 2022; Li et al., 2023a; Lin et al., 2024), or global Lipschitz smoothness in (Lee et al., 2024).[6] This difference in the dynamics assumption reflects a *trade-off* with the disturbance assumptions:

---

[5]A function $\gamma : \mathbb{R}_{\geq 0} \to \mathbb{R}_{\geq 0}$ is a $\mathcal{K}$ function if it is continuous, strictly increasing and $\gamma(0) = 0$. A function $\beta : \mathbb{R}_{\geq 0} \times \mathbb{R}_{\geq 0} \to \mathbb{R}_{\geq 0}$ is a $\mathcal{KL}$ function if, for each fixed $t \geq 0$, the function $\beta(\cdot, t)$ is a $\mathcal{K}$ function, and for each fixed $s \geq 0$, the function $\beta(s, \cdot)$ is decreasing and $\beta(s, t) \to 0$ as $t \to \infty$.

[6]Global Lipschitz smoothness may exclude system dynamics with higher-order polynomials.

we assume a stronger assumption on the boundedness of disturbances and a weaker assumption on local stability, whereas much of the literature considers (sub)Gaussian distributions (which can be unbounded) but requires stronger global properties for dynamics. Since this paper is largely motivated by physical systems, which typically encounter bounded disturbances/inputs and generally only satisfy local stability (Slotine and Li, 1991), we address this trade-off through our current set of assumptions, leaving it as an exciting future direction to consider relaxing these assumptions.

## 3 Main Results

In this section, we provide the estimation error bounds of LSE for linearly parameterized nonlinear systems under i.i.d. random inputs. The estimation error bounds rely on the establishment of probabilistic persistent excitation, which will be introduced in the first subsection. Later, we also generalize the results to include control policies and discuss the convergence rate of another popular uncertainty quantification method in the control literature, set membership estimation, whose formal definition is deferred to the corresponding subsection.

### 3.1 Probabilistic Persistent Excitation

It is well-known that persistent excitation (PE) is a crucial condition for successful system identification (Narendra and Annaswamy, 1987). In the following, we introduce the persistent excitation condition for our linearly parameterized nonlinear systems.

**Definition 3** (Persistent excitation (Skantze et al., 2000; Sastry and Bodson, 2011)). *System* (1) *is persistently excited if there exist $s > 0$ and $m \geq 1$ such that for any $t_0 \geq 0$, we have*

$$\frac{1}{m} \sum_{t=t_0}^{t_0+m-1} \phi(x_t, u_t)\phi^{\mathsf{T}}(x_t, u_t) \succeq s^2 I_{n_\phi}.$$

In the stochastic setting, PE is closely related with a block-martingale small-ball (BMSB) condition proposed in Simchowitz et al. (2018), which can be viewed as a probabilistic version of PE.

**Definition 4** (BMSB (Simchowitz et al., 2018)). *Let $\{\mathcal{F}_t\}_{t \geq 1}$ denote a filtration and let $\{y_t\}_{t \geq 1}$ be an $\{\mathcal{F}_t\}_{t \geq 1}$-adapted random process taking values in $\mathbb{R}^{n_y}$. We say $\{y_t\}_{t \geq 1}$ satisfies the $(k, \Gamma_{sb}, p)$-block martingale small-ball (BMSB) condition for a positive integer $k$, a $\Gamma_{sb} \succ 0$, and a $p \in [0, 1]$, if for any fixed $v \in \mathbb{R}^{n_y}$ such that $\|v\|_2 = 1$, the process $\{y_t\}_{t \geq 1}$ satisfies $\frac{1}{k}\sum_{i=1}^{k} \mathbb{P}(|v^{\mathsf{T}} y_{t+i}| \geq \sqrt{v^{\mathsf{T}}\Gamma_{sb}v} \mid \mathcal{F}_t) \geq p$ almost surely for any $t \geq 1$.*

One major contribution of this paper is formally establishing the BMSB condition for linearly parameterized nonlinear systems with real-analytic feature functions.

In the following, we first investigate the open-loop system with i.i.d. inputs and later extend the results to the closed-loop systems with inputs $u_t = \pi(x_t) + \eta_t$, where $\eta_t$ represents the noise and $\pi : \mathbb{R}^{n_x} \to \mathbb{R}^{n_u}$ denotes a control policy. The following theorem considers the open-loop systems.

**Theorem 1** (BMSB for open-loop systems). *Let $u_t = \eta_t$ and consider the filtration $\mathcal{F}_t = \mathcal{F}(w_0, \cdots, w_{t-1}, x_0, \cdots, x_t, \eta_0, \cdots, \eta_t)$. Suppose Assumptions 1, 2, 3, 4 hold, then there exist $s_\phi > 0$ and $p_\phi \in (0, 1)$ such that the $\{\mathcal{F}_t\}_{t \geq 1}$-adapted process $\{\phi(x_t, u_t)\}_{t \geq 1}$ satisfies the $(1, s_\phi^2 I_{n_\phi}, p_\phi)$-BMSB condition.*

*Proof Sketch.* Intuitively, BMSB requires that any linear combination of feature functions remains positive with a non-vanishing probability. Notice that a linear combination of real-analytic functions is itself real-analytic, and the zeros of an analytic function have measure zero. These facts allow us to show that the probability of a linear combination of linearly independent feature functions equaling zero is less than one, as long as the noises follow semi-continuous distributions, by the connection of the Lebesgue measure and the probability measure in Definition 1.

In more detail, the proof leverages a variant of the Paley-Zygmund argument (Petrov, 2007), which provides a lower bound for the tail properties of positive random variables. Specifically, it states that the probability of a positive random variable being small depends on the ratio of its even moments.

We apply this result to the random variable $|v^T\phi(x_{t+1}, u_{t+1}) \mid \mathcal{F}_t|$ with $\|v\|_2 = 1$ and aim to show that the lower bound is non-trivial for any direction $v$ with $\|v\|_2 = 1$ and any filtration $\mathcal{F}_t$, $t \geq 0$. We then use results from measure theory to demonstrate the existence of such a non-trivial lower bound. This is done by showing that the Lebesgue measure of the set where $|v^T\phi(x_{t+1}, u_{t+1})| = 0$ is zero, and thus the even moments of $|v^T\phi(x_{t+1}, u_{t+1}) \mid \mathcal{F}_t|$ are non-zero, provided that the noise and disturbance distributions are semi-continuous. For further details, please refer to Appendix A. □

It is worth pointing out that Theorem 1 only establishes the existence of the constants $(s_\phi, p_\phi)$, and deriving explicit formulas of these constants are left for future work. In particular, it can be challenging to derive a generic formula for all linearly parameterized nonlinear systems, but an exciting direction is to study reasonable sub-classes of systems and construct their corresponding formulas of the constants $(s_\phi, p_\phi)$.

## 3.2 Non-asymptotic Bounds for LSE

We are now prepared to present the non-asymptotic convergence rate for the LSE in learning the unknown parameters of the system (1).

**Theorem 2** (LSE's convergence rate for open-loop systems). *Consider the dynamical system described in (1) with i.i.d. inputs $u_t = \eta_t$ and assume that Assumptions 1, 2, 3, 4 are satisfied. Let $s_\phi$ and $p_\phi$ be as defined in Theorem 1, and define $\bar{b}_\phi = \sup_{t\geq 0} \mathbb{E}\big[\|\phi(z_t)\|_2^2\big]$. For a fixed $\delta \in (0, 1)$ and $T \geq 1$, if $T$ satisfies the condition*

$$T \geq \frac{10}{p_\phi}\left(\log\left(\frac{1}{\delta}\right) + 2n_\phi \log\left(\frac{10}{p_\phi}\right) + n_\phi \log\left(\frac{\bar{b}_\phi}{\delta s_\phi^2}\right)\right),$$

*then LSE's estimation $\hat{\theta}_T$ satisfies the following error bound with probability at least $1 - 3\delta$.*

$$\|\hat{\theta}_T - \theta_*\|_2 \leq \frac{90\sigma_w}{p_\phi}\sqrt{\frac{n_x + \log\left(\frac{1}{\delta}\right) + n_\phi \log\left(\frac{10}{p_\phi}\right) + n_\phi \log\left(\frac{\bar{b}_\phi}{\delta s_\phi^2}\right)}{T s_\phi^2}}.$$

The proof relies on Theorem 1 and Theorem 2.4 in (Simchowitz et al., 2018). The complete proof is provided in Appendix B.1.

Theorem 2 demonstrates that LSE converges to the true parameters under random control inputs and random disturbances (non-active exploration) at a rate of $\frac{1}{\sqrt{T}}$ for linearly parameterized nonlinear systems. This is consistent with the convergence rates of LSE for linear systems in terms of $T$.

Regarding the dependence of the convergence rate on dimension, the explicit dependence is $\sqrt{n_x + n_\phi}$, where $n_x$ and $n_\phi$ refer to the dimensions of the state and the characteristic vector, respectively. In addition, it is worth mentioning that other parameters, such as $s_\phi, p_\phi, \bar{b}_\phi$, may also implicitly depend on the dimensions. For some special systems, such as bilinear systems, it has been shown that these constants are independent of dimensions (Sattar et al., 2022). It is left as future work to explore other nonlinear systems' implicit dimension dependence.

Next, we can generalize the i.i.d. inputs $u_t$ to include control policies, i.e., $u_t = \pi(x_t) + \eta_t$, where $\eta_t$ satisfies Assumption 3 and $\pi(x_t)$ is analytic.

**Corollary 1** (LSE's convergence rate for closed-loop systems). *Consider inputs $u_t = \pi(x_t) + \eta_t$, where $\pi(\cdot)$ is real-analytic, $\eta_t$ satisfies Assumption 3, and the closed-loop system $x_{t+1} = \theta_*\phi(x_t, \pi(x_t) + \eta_t) + w_t$ satisfies Assumption 4 for both $w_t$ and $\eta_t$. Then, the same convergence rate as in Theorem 2 holds.*

The proof is provided in Appendix B.2.

## 3.3 Non-asymptotic Diameter Bounds for SME

Set membership estimation (SME) is another popular method for uncertainty quantification in control system estimation (Bertsekas, 1971; Fogel and Huang, 1982; Lu and Cannon, 2023; Li et al., 2024). Unlike LSE, SME is a set-estimator and directly estimates the uncertainty set. Since the analysis of SME also relies on the probabilistic persistent excitation analysis, we can also establish the

convergence rate of SME for linearly parameterized nonlinear systems under i.i.d. noises in the following. In particular, SME estimates the uncertainty set as

$$\Theta_T = \bigcap_{t=0}^{T-1} \left\{ \hat{\theta} : x_{t+1} - \hat{\theta}\phi(x_t, u_t) \in \mathcal{W} \right\}, \tag{3}$$

where $\mathcal{W}$ is a bounded set such that $w_t \in \mathcal{W}$ for all $t \geq 0$.

The convergence of SME relies on an additional assumption as shown in the following: the tightness of the bound $\mathcal{W}$ on the support of $w_t$'. This tightness assumption is commonly considered in SME's literature (Li et al., 2024; Lu et al., 2019; Akçay, 2004). In addition, (Li et al., 2024) discusses the relaxation of this assumption by learning a tight bound at the same time as learning the uncertainty set of $\theta_*$ for linear systems. Similar tricks can be applied to nonlinear systems, but this paper only considers the vanilla case of SME for simplicity.

**Assumption 5** (Tight bound on disturbances). *Assume for any $\epsilon > 0$, there exists $q_w(\epsilon) > 0$, such that for any $1 \leq j \leq n$ and $t \geq 0$, we have $\mathbb{P}(w_t^j + w_{\max} \leq \epsilon) \geq q_w(\epsilon) > 0$, $\mathbb{P}(w_{\max} - w_t^j \leq \epsilon) \geq q_w(\epsilon) > 0$.*

Assumption 5 requires that $w_t$ can visit the boundary of the set $\mathcal{W}$' arbitrarily closely with a positive probability. For example, for a one-dimensional $w_t$ bounded by $-w_{\max} \leq w_t \leq w_{\max}$, the assumption 5 requires that there is a positive probability that $w_t$ is close to $w_{\max}$ and $-w_{\max}$, that is, for any $\epsilon > 0$, we have $\mathbb{P}(w_{\max} - \epsilon \leq w_t \leq w_{\max}) > 0$ and $\mathbb{P}(-w_{\max} \leq w_t \leq -w_{\max} + \epsilon) > 0$.

Next, we state a non-asymptotic bound on the diameter of the uncertainty set estimated by SME.

**Theorem 3** (SME's diameter bound for open-loop systems). *Consider the system (1) with i.i.d. inputs $u_t = \eta_t$. Suppose Assumptions 1, 2, 3, 4 are satisfied. Consider $s_\phi$ and $p_\phi$ defined in Theorem 1 and let $b_\phi = \sup_{t \geq 0} \|\phi(z_t)\|_2$. For any $m \geq 0$ and $\delta \in (0, 1)$, when $T > m$, we have*

$$\mathbb{P}\left( \text{diam}(\Theta_T) > \delta \right) \leq \frac{T}{m} \tilde{O}\left(n_\phi^{2.5}\right) a_2^{n_\phi} \exp(-a_3 m) \; + \; \tilde{O}\left(n_x^{2.5} n_\phi^{2.5}\right) a_4^{n_x n_\phi} \left( 1 - q_w\left( \frac{a_1 \delta}{4\sqrt{n_x}} \right) \right)^{\frac{T}{m}},$$

*where $a_1 = \frac{s_\phi p_\phi}{4}$, $a_2 = \frac{64 b_\phi^2}{s_\phi^2 p_\phi^2}$, $a_3 = \frac{p_\phi^2}{8}$, $a_4 = \frac{16 b_\phi \sqrt{n_x}}{s_\phi p_\phi}$. The constants hidden in $\tilde{O}$ are provided in the Appendix C.1.*

The proof of Theorem 3 is based on Theorem 1 in this paper and Theorem 1 from (Li et al., 2024). The detailed proof is provided in Appendix C.1.

Theorem 3 establishes an upper bound on the "failure" probability of SME, that is, the probability that the diameter of the uncertainty set exceeds $\delta$. To ensure that the failure probability is less than 1, one can select $m = O(\log(T))$ and choose a sufficiently large $T$ such that $T \geq m = O(\log(T))$. If $w_t$ is more likely to visit the boundaries of the set $\mathcal{W}$ (meaning larger $q(\ell)$), the SME is less likely to estimate an uncertainty set with a diameter greater than $\delta$.

To provide more intuitions on the diameter bound in Theorem 3, we consider $q_w(\ell) = c_w \ell$ for some $c_w > 0$. Note that several common distributions, including the uniform distribution and the truncated Gaussian distribution, satisfy this property on $q_w(\ell)$ (see Appendix C.2 for explicit formulas of $c_w$). With $q_w(\ell) = c_w \ell$, we can provide a convergence rate of the SME in terms of $T$ in the following.

**Corollary 2** (SME's convergence rate when $q_w(\ell) = c_w \ell$). *For any $\epsilon > 0$, let*

$$m \geq O\left( \frac{\log\left(\frac{T}{\epsilon}\right) + n_\phi \log\left(\frac{8 b_\phi}{s_\phi p_\phi}\right)}{p_\phi^2} \right).$$

*If $w_t$'s distribution satisfies $q_w(\ell) = c_w \ell$ for all $\ell > 0$, then when $T > m$, we have:*

$$\text{diam}(\Theta_T) \leq O\left( \frac{m\sqrt{n_x} \log\left(\frac{1}{\epsilon}\right) + m n_x^{1.5} n_\phi \log\left(\frac{b_\phi n_x}{s_\phi p_\phi}\right)}{c_w s_\phi p_\phi T} \right),$$

*with probability at least $1 - 2\epsilon$. Here the constants hidden in $O(\cdot)$ are provided in Appendix C.*

The proof of Corollary 2 is provided in Appendix C.2.

Finally, similar to LSE, we can extend SME's convergence rates from open-loop systems to closed-loop systems with real-analytic control policies, i.e., $u_t = \pi(x_t) + \eta_t$, where $\eta_t$ satisfies Assumption 3 and $\pi(x_t)$ is real-analytic.

**Corollary 3** (SME's convergence rate for closed-loop systems). *Consider inputs* $u_t = \pi(x_t) + \eta_t$, *where* $\pi(\cdot)$ *is real-analytic,* $\eta_t$ *satisfies Assumption 3, and the closed-loop system* $x_{t+1} = \theta_* \phi(x_t, \pi(x_t) + \eta_t) + w_t$ *satisfies Assumption 4 in terms of both* $w_t$ *and* $\eta_t$. *Then, the same convergence rates as in Theorem 3 and Corollary 2 still hold.*

The proof of Corollary 3 is provided in Appendix C.3.

# 4    Numerical Experiments

In this section, we evaluate the performance of LSE in estimating the unknown parameter $\theta_*$ and SME in estimating the uncertainty set for the unknown parameters using the pendulum and quadrotor examples outlined in Section 2. We compare the empirical convergence rates of LSE and SME with the theoretical rates in Theorem 2 and Corollary 2. In each case, the input $u_t$ is composed of a control policy and i.i.d. noise, such that $u_t = \pi(x_t) + \eta_t$. For our experiments, we employ noise and disturbances drawn from uniform and truncated-Gaussian distributions. To compute theoretical rates, we numerically estimate parameters such as $s_\phi$ and $p_\phi$ (see Appendix E). Further details can be found in our source code.[7]

The details for these scenarios are outlined below:

- **Pendulum 1:** In the pendulum example described in Section 2, the control input is $u_t = -k\dot{\alpha}_t + \eta_t$. This scenario includes two unknown parameters: $\theta_1 = \dfrac{1}{l}$ and $\theta_2 = \dfrac{1}{ml^2}$.

- **Quadrotor 2:** For the quadrotor example in Section 2, the control input is defined as $u_t = \pi(x_t) + \eta_t$, where $\pi(x_t)$ follows the controller proposed by Alaimo et al. (2013). The quadrotor system involves 13 states and 4 inputs, with the unknown parameter matrix $\theta_*$ containing 7 parameters, including the mass $m$ and specific elements of the inertia matrix $I$.

Further details on controller gains and unknown parameters are provided in Appendix D.

**LSE Results:**    Figures 1a and 1b present a comparison between the LSE theoretical bound from Theorem 2 with its empirical estimation error of the unknown parameters $\theta_*$ versus trajectory length $T$ for the pendulum example, with uniform and truncated-Gaussian noises and disturbances. Similarly, Figures 1c and 1d show this comparison for the quadrotor example. In each figure, both the theoretical bound and empirical error are normalized by the $l_2$ norm of the nominal parameter $\theta_*$. The log-log plots for both scenarios demonstrate that the empirical error rate achieves $O(\frac{1}{\sqrt{T}})$ which in consistent with the theoretical rate in Theorem 2.

**SME Results:**    Figures 2a and 2b show the empirical convergence rate of SME for the pendulum example, for uniform and truncated-Gaussian noises and disturbances, in comparison to the theoretical rate from Corollary 2. Both the theoretical bound and empirical error in the figures are normalized by the $l_2$ norm of the nominal parameter $\theta_*$. The log-log plots indicate that the empirical rate achieves $O(\frac{1}{T})$, which is consistent with the results from Corollary 2 and with the related results for linear systems in (Li et al., 2024). A similar result can be observed for the quadrotor example, in Figures 2c and 2d. Additionally, Figure 3 shows the uncertainty sets estimated by SME for the two unknown parameters, labeled $\theta_1$ and $\theta_2$, in the pendulum example, along with the diameters of these sets as trajectory length grows. We observe that these sets contract as trajectory length increases, with the true values of the unknown parameters lying within the estimated uncertainty sets. The illustration of uncertainty sets for the quadrotor example is provided in Appendix D.2.

---

[7]`https://github.com/NeginMusavi/real-analytic-nonlinear-sys-id`

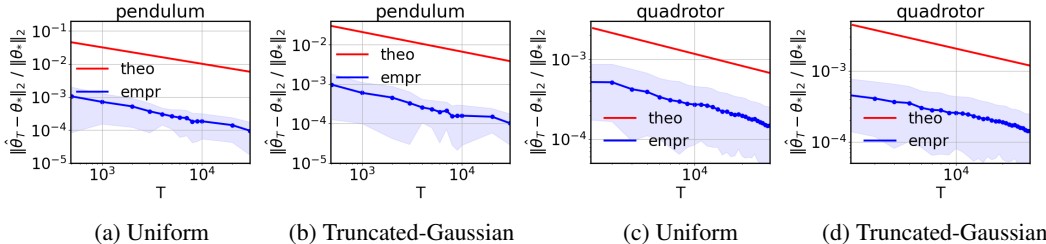

Figure 1: Convergence rate of the LSE for pendulum and quadrotor scenarios: (a) Pendulum example with uniform, (b) Pendulum example with truncated-Gaussian, (c) Quadrotor example with uniform, and (d) Quadrotor example with truncated-Gaussian noises and disturbances. Here, uniform noises and disturbances are i.i.d. generated from $\texttt{uniform}([-1, 1])$, and truncated-Gaussian noises and disturbances are i.i.d. generated from $\texttt{truncated-Gaussian}(0, 0.1, [-1, 1])$. "theo" denotes the theoretical convergence rate, and "empr" represents the empirical rate. The mean error across 20 trials is shown by dots on the empirical plots, with shaded areas illustrating empirical standard deviation.

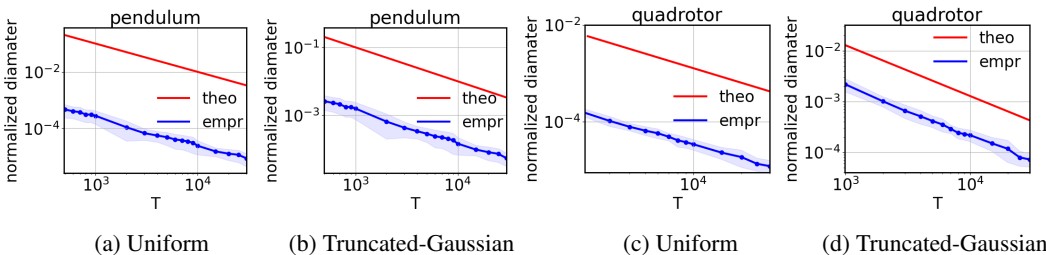

Figure 2: Convergence rate of the SME for pendulum and quadrotor scenarios: (a) Pendulum example with uniform, (b) Pendulum example with truncated-Gaussian, (c) Quadrotor example with uniform, and (d) Quadrotor example with truncated-Gaussian noises and disturbances. Here, uniform noises and disturbances are i.i.d. generated from $\texttt{uniform}([-1, 1])$, and truncated-Gaussian noises and disturbances are i.i.d. generated from $\texttt{truncated-Gaussian}(0, 0.5, [-1, 1])$. "theo" denotes the theoretical convergence rate, and "empr" represents the empirical rate. The mean error across 10 trials is shown by dots on the empirical plots, with shaded areas illustrating empirical standard deviation.

## 5   Concluding Remarks

**Conclusion.** This study examines the probabilistic persistent excitation in a class of nonlinear systems influenced by i.i.d. noise and stochastic disturbances, with the stipulation that their distributions do not concentrate on sets of Lebesgue measure zero. Based on this we then present an explicit bound on the convergence rate of SME estimations and LSE estimations for this class of dynamical systems. Additionally, numerical experiments in the context of robotics are provided to illustrate both methods.

**Limitations.** One limitation of this work is that our analysis relies on a specific class of i.i.d. noises and stochastic disturbances, where the probability distribution is not concentrated on sets of Lebesgue measure zero. While this is a sufficient condition, it is possible that the BMSB conditions are satisfied under other circumstances. Another limitation is that, though we provide sufficient conditions for the existence of parameters satisfying the BMSB condition, the explicit dependence is not detailed here. Lastly, imperfect observations are not considered here.

**Future Work.** Our future work includes several promising directions, e.g., to explore cases that do not satisfy our semi-continuity assumption, such as discrete noises, and to investigate the explicit dependence of the BMSB parameter on system attributes, such as state, input, and feature dimensions, etc. Furthermore, extending this work to imperfect state observations is an important next step. Finally, a potential direction is to provide a non-asymptotic analysis of the volumes of uncertainty sets estimated by SME uncertainty sets, as opposed to the current focus on their diameters.

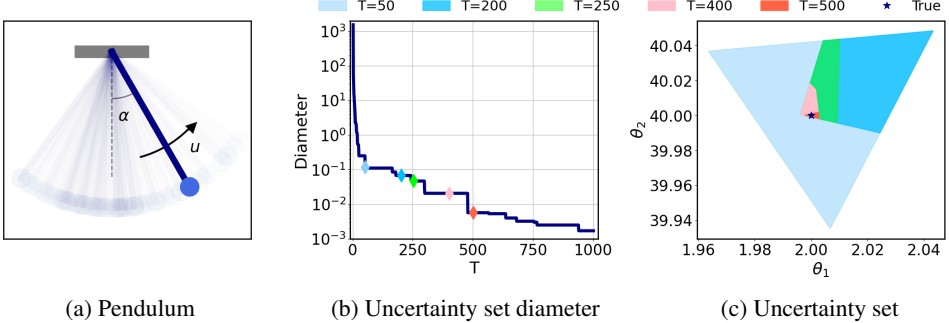

| (a) Pendulum | (b) Uncertainty set diameter | (c) Uncertainty set |

Figure 3: Performance of SME for pendulum in (a) with control input $u_t = -k\dot{\alpha}_t + \eta_t$ where $k = 0.1$, $\eta_t$ i.i.d. generated from `truncated-Gaussian`$(0, 2, [-2, 2])$ and disturbed with $w_t$ i.i.d. generated from `truncated-Gaussian`$(0, 1, [-1, 1])$. (b) Diameter of the uncertainty set estimated by SME. (c) Uncertainty set depicted for $T = 50, 200, 250, 400, 500$.

## Broader Impact

This paper is a foundation research and develops theoretical insight to estimation of nonlinear control systems. We do not see a direct path to negative applications in general. But we want to mention that successful applications of our theoretical results rely on verifying the assumptions in this paper.

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

# Appendix

## Roadmap

- Appendix A provides a proof of Theorem 1.

- Appendix B provides proofs of Theorem 2 and Corollary 1.

- Appendix C presents a proof of Theorem 3 and Corollaries 2 and 3.

- Appendix D provides more details of the simulation settings.

- Appendix E discusses the numerical estimation of the BMSB parameters $(s_\phi, p_\phi)$ in Theorem 1.

- The NeurIPS Paper Checklist is provided after the appendices.

## A    Proof Theorem 1

*Proof.* Given that $u_t = \eta_t$ and satisfies the conditions in Assumption 3, $u_t$ is bounded, meaning $u_t \in \mathcal{U}$, where $\mathcal{U}$ is a compact set. Moreover, since the system is LISS, there exist functions $\gamma \in \mathcal{K}$ and $\beta \in \mathcal{KL}$, such that for all $t \geq 0$, the following holds:

$$x_t \in \mathcal{X} = \left\{ x \in \mathbb{R}^n : \|x\|_2 \leq \beta(\rho_x, 0) + \gamma(\rho) \right\}$$

with parameters $\rho_x \geq \|x_0\|_2$ and $\rho \geq \max(w_{\max}, u_{\max})$. Let $\mathcal{Z} = \mathcal{X} \times \mathcal{U}$, then $z_t \in \mathcal{Z}$ for all $t \geq 0$. The set $\mathcal{Z}$ is a compact subset of $\mathbb{R}^{n_x + n_u}$.

To show that the $\{\mathcal{F}_t\}_{t \geq 1}$-adapted process $\{\phi(z_t)\}_{t \geq 1}$ satisfies the BMSB condition, it is sufficient to demonstrate that there exist $s_\phi > 0$ and $p_\phi \in (0, 1)$ such that for all $t \geq 0$ and for any $v \in \mathbb{R}^{n_\phi}$ with $\|v\|_2 = 1$, the following holds:

$$\mathbb{P}\left( |v^T \phi(z_{t+1})| \geq s_\phi \|v\|_2 \mid \mathcal{F}_t \right) \geq p_\phi. \tag{4}$$

To establish this, we apply the Paley-Zygmund inequality, which gives a lower bound on the tail probability of a non-negative random variable:

**Lemma 1.** *(Paley-Zygmund (Petrov, 2007)) Let $x$ be a non-negative random variable. Then for any $r \in (0, 1)$, the following holds:*

$$\mathbb{P}\left( x > r\sqrt{\mathbb{E}[x^2]} \right) \geq (1 - r^2)^2 \frac{\mathbb{E}[x^2]^2}{\mathbb{E}[x^4]}.$$

Based on this result, for any $r \in (0, 1)$, we have:

$$\mathbb{P}\left( |v^\mathsf{T} \phi(z_{t+1})| > r\sqrt{\mathbb{E}\left[ \left(v^\mathsf{T}\phi(z_{t+1})\right)^2 \mid \mathcal{F}_t \right]} \;\middle|\; \mathcal{F}_t \right) \geq (1 - r^2)^2 \frac{\mathbb{E}\left[ \left(v^\mathsf{T}\phi(z_{t+1})\right)^2 \mid \mathcal{F}_t \right]^2}{\mathbb{E}\left[ \left(v^\mathsf{T}\phi(z_{t+1})\right)^4 \mid \mathcal{F}_t \right]}. \tag{5}$$

Let $\mathcal{V} = \{v \in \mathbb{R}^{n_\phi} : \|v\|_2 = 1\}$. To show that the BMSB condition holds, it is sufficient to establish the following two points:

- $\displaystyle \inf_{\mathcal{F}_t,\, t \geq 0} \inf_{v \in \mathcal{V}} \mathbb{E}\left[ \left(v^\mathsf{T}\phi(z_{t+1})\right)^2 \mid \mathcal{F}_t \right] > 0,$

- and $\displaystyle \sup_{\mathcal{F}_t,\, t \geq 0} \sup_{v \in \mathcal{V}} \mathbb{E}\left[ \left(v^\mathsf{T}\phi(z_{t+1})\right)^4 \mid \mathcal{F}_t \right] < \infty.$

These conditions ensure that the $\{\mathcal{F}_t\}_{t\geq 1}$-adapted process $\{\phi(z_t)\}_{t\geq 1}$ satisfies the BMSB condition with some constants $s_\phi > 0$ and $p_\phi \in (0,1)$. We will divide the proof into two parts:

**Step 1. Showing that** $\displaystyle \inf_{\mathcal{F}_t,\, t\geq 0} \inf_{v\in\mathcal{V}} \mathbb{E}\big[\big(v^\intercal\phi(z_{t+1})\big)^2 \mid \mathcal{F}_t\big] > 0$**:**

We begin by noting the following:

$$\inf_{\mathcal{F}_t,\, t\geq 0} \inf_{v\in\mathcal{V}} \mathbb{E}\left[\big(v^\intercal\phi(z_{t+1})\big)^2 \mid \mathcal{F}_t\right] = \inf_{\mathcal{F}_t,\, t\geq 0} \inf_{v\in\mathcal{V}} \mathbb{E}\left[\big(v^\intercal\phi(x_{t+1},u_{t+1})\big)^2 \mid \mathcal{F}_t\right]$$

$$= \inf_{\mathcal{F}_t,\, t\geq 0} \inf_{v\in\mathcal{V}} \mathbb{E}\left[\big(v^\intercal\phi\big(\theta_*\phi(z_t) + w_t, u_{t+1}\big)\big)^2 \mid \mathcal{F}_t\right].$$

Since $z_t \in \mathcal{F}_t$ while $w_t, u_{t+1} \notin \mathcal{F}_t$, we can treat $z_t$ as a constant and $w_t, u_{t+1} = \eta_{t+1}$ as random variables. From the continuity of features $\phi(\cdot)$, we can conclude that:

$$\inf_{\mathcal{F}_t,\, t\geq 0} \inf_{v\in\mathcal{V}} \mathbb{E}\left[\big(v^\intercal\phi(z_{t+1})\big)^2 \mid \mathcal{F}_t\right] = \inf_{z\in\mathcal{Z}} \inf_{v\in\mathcal{V}} \mathbb{E}\left[\big(v^\intercal\phi\big(\underbrace{\theta_*\phi(z) + w}_{=:\, h(z,w)}, \eta\big)\big)^2\right],$$

where $w, \eta$ are independent random variables, as assumed in Assumptions 2 and 3. Now, let $\mathcal{N}_v^z = \big\{(w,\eta) \in \mathcal{W}\times\mathcal{U} : v^\intercal\phi\big(h(z,w),\eta\big) = 0\big\}$, and we have:

$$\mathbb{E}\left[\big(v^\intercal\phi\big(h(z,w),\eta\big)\big)^2\right] = \underbrace{\mathbb{E}\left[\big(v^\intercal\phi\big(h(z,w),\eta\big)\big)^2 \mathbb{1}\big\{v^\intercal\phi\big(h(z,w),\eta\big) = 0\big\}\right]}_{=0}$$

$$+ \mathbb{E}\left[\big(v^\intercal\phi\big(h(z,w),\eta\big)\big)^2 \mathbb{1}\big\{v^\intercal\phi\big(h(z,w),\eta\big) \neq 0\big\}\right]$$

$$= \mathbb{E}\left[\big(v^\intercal\phi\big(h(z,w),\eta\big)\big)^2 \mid (w,\eta) \notin \mathcal{N}_v^z\right]\mathbb{P}\Big((w,\eta) \notin \mathcal{N}_v^z\Big)$$

$$= \mathbb{E}\left[\big(v^\intercal\phi\big(h(z,w),\eta\big)\big)^2 \mid (w,\eta) \notin \mathcal{N}_v^z\right]\Big(1 - \mathbb{P}\Big((w,\eta) \in \mathcal{N}_v^z\Big)\Big).$$

Therefore, we have:

$$\inf_{z\in\mathcal{Z}} \inf_{v\in\mathcal{V}} \mathbb{E}\left[\big(v^\intercal\phi\big(h(z,w),\eta\big)\big)^2\right] = \inf_{z\in\mathcal{Z}} \inf_{v\in\mathcal{V}} \mathbb{E}\left[\big(v^\intercal\phi\big(h(z,w),\eta\big)\big)^2 \mid (w,\eta) \notin \mathcal{N}_v^z\right]$$
$$\times \left(1 - \sup_{z\in\mathcal{Z}} \sup_{v\in\mathcal{V}} \mathbb{P}\Big((w,\eta) \in \mathcal{N}_v^z\Big)\right). \tag{6}$$

It is evident that if $\mathcal{N}_v^z = \emptyset$, then

$$\inf_{z\in\mathcal{Z}} \inf_{v\in\mathcal{V}} \mathbb{E}\left[\big(v^\intercal\phi\big(h(z,w),\eta\big)\big)^2\right] \neq 0, \text{ and } \sup_{z\in\mathcal{Z}} \sup_{v\in\mathcal{V}} \mathbb{P}\Big((w,\eta) \in \mathcal{N}_v^z\Big) = 0,$$

leading to $\displaystyle \inf_{z\in\mathcal{Z}} \inf_{v\in\mathcal{V}} \mathbb{E}\left[\big(v^\intercal\phi\big(h(z,w),\eta\big)\big)^2\right] > 0$. Now we proceed with the case where $\mathcal{N}_v^z \neq \emptyset$. For this, we can use the following lemma concerning the zero set of real-analytic functions in terms of Lebesgue measure.

**Lemma 2** (The zero set of real-analytic functions (Crăciun and Ghoshdastidar, 2024)). *The set of zeros of a non-trivial real-analytic function $f : \mathbb{R}^n \to \mathbb{R}$ has a Lebesgue measure zero in $\mathbb{R}^n$.*

This is a known result and can be proved using the identity theorem along with Fubini's theorem. For further information on this topic, see sources such as (Krantz and Parks, 2002; Bogachev and Ruas, 2007).

Recall that we defined $h(z,w) = \theta_*\phi(z) + w$. Notice that $h(\cdot,\cdot)$ is real-analytic. Now consider

$$v^\intercal\phi\big(h(z,w),\eta\big) = \sum_{i=1}^{n_\phi} v^i\phi^i\big(h(z,w),\eta\big),$$

where $\phi^i\big(h(z,w),\eta\big)$ are linearly independent. Hence, the sum $\sum_{i=1}^{n_\phi} v^i \phi^i\big(h(z,w),\eta\big) \not\equiv 0$ for any $v \in \mathcal{V}$. This implies that $v^\intercal \phi\big(h(z,w),\eta\big)$ is real-analytic and non-zero. Consequently, by Lemma 2, $\lambda^{n_x+n_u}(\mathcal{N}_v^z) = 0$ for any $v \in \mathcal{V}$.

Under Assumptions 3 and 2, there cannot exist a set $\mathcal{E} \subset \mathcal{Z}$ of Lebesgue measure zero in $\mathbb{R}^{n_x+n_u}$ for which the $\mathbb{P}\big((w,\eta) \in \mathcal{E}\big) = 1$. Taking this into account, along with the fact that $\lambda^{n_x+n_u}(\mathcal{N}_v^z) = 0$ and that the sets $\mathcal{V}$ and $\mathcal{Z}$ are closed sets (implying they include all their limit points), we can conclude that

$$\sup_{z \in \mathcal{Z}} \sup_{v \in \mathcal{V}} \mathbb{P}\bigg((w,\eta) \in \mathcal{N}_v^z\bigg) \neq 1.$$

Moreover, since $\lambda^{n_x+n_u}(\mathcal{N}_v^z) = 0$ and $\lambda^{n_x+n_u}(\mathcal{W} \times \mathcal{U}) \neq 0$, it follows that $(\mathcal{N}_v^z)^c \neq \emptyset$. This implies

$$\inf_{z \in \mathcal{Z}} \inf_{v \in \mathcal{V}} \mathbb{E}\bigg[\big(v^\intercal \phi\big(h(z,w),\eta\big)\big)^2 \mid (w,\eta) \notin \mathcal{N}_v^z\bigg] \neq 0.$$

Substituting these results into (6), we obtain:

$$\inf_{\mathcal{F}_t,\ t\geq 0} \inf_{v \in \mathcal{V}} \mathbb{E}\big[\big(v^\intercal \phi(z_{t+1})\big)^2 \mid \mathcal{F}_t\big] > 0.$$

**Step 2. Showing that** $\displaystyle\sup_{\mathcal{F}_t,\ t\geq 0} \sup_{v \in \mathcal{V}} \mathbb{E}\big[\big(v^\intercal \phi(z_{t+1})\big)^4 \mid \mathcal{F}_t\big] < \infty$**:**

Since $z_t \in \mathcal{Z}$ for $t \geq 0$, and considering that the noise and disturbances are bounded while the features are real-analytic, it follows that $z_{t+1}|\mathcal{F}_t$ is a bounded random variable. Consequently, $v^\intercal \phi(z_{t+1})|\mathcal{F}_t$ is also bounded. Given that both $\mathcal{Z}$ and $\mathcal{V}$ are compact sets—meaning they contain all their limit points—and that any random variable with bounded support has finite moments, we conclude that

$$\sup_{\mathcal{F}_t,\ t\geq 0} \sup_{v \in \mathcal{V}} \mathbb{E}\big[\big(v^\intercal \phi(z_{t+1})\big)^4 \mid \mathcal{F}_t\big] < \infty.$$

We finalize the proof by combining the results from Step 1 and Step 2. $\qquad\qquad\square$

# B  Proofs for Theorem 2 and Corollary 1

## B.1  Proof of Theorem 2

*Proof.* The proof hinges on the following key meta-theorem about the LSE convergence rate:

**Theorem 4** (LSE meta-theorem (Simchowitz et al., 2018))**.** *Fix $\delta \in (0,1)$, $T \geq 1$, and $0 \prec \Gamma_{sb} \prec \bar{\Gamma}$. Consider a random process $\{(y_t, x_t)\}_{t\geq 1} \in (\mathbb{R}^{n_y} \times \mathbb{R}^{n_x})^T$, and a filtration $\{\mathcal{F}_t\}_{t\geq 1}$. Suppose the following conditions hold:*

- *$x_t = \theta_* y_t + w_t$, where $w_t|\mathcal{F}_t$ is a zero mean $\sigma_w^2$-sub-Gaussian,*

- *$\{y_t\}_{t\geq 1}$ is an $\{\mathcal{F}_t\}_{t\geq 1}$-adapted random process satisfying the $(k, \Gamma_{sb}, p)$-BMSB condition,*

- *$\mathbb{P}\big(\sum_{t=1}^{T} y_t y_t^\intercal \not\succeq T\bar{\Gamma}\big) \leq \delta$.*

*If the trajectory length $T$ satisfies*

$$T \geq \frac{10k}{p^2}\left(\log\left(\frac{1}{\delta}\right) + \log\det\left(\bar{\Gamma}\Gamma_{sb}^{-1}\right) + 2n_y \log\left(\frac{10}{p}\right)\right),$$

*then with probability at least $1 - 3\delta$, LSE estimation error is bounded by:*

$$\big\|\hat{\theta}_T - \theta_*\big\|_2 \leq \frac{90\sigma_w}{p}\sqrt{\frac{n_x + \log\left(\frac{1}{\delta}\right) + \log\det\left(\bar{\Gamma}\Gamma_{sb}^{-1}\right) + n_y \log\left(\frac{10}{p}\right)}{T\sigma_{min}(\Gamma_{sb})}}.$$

Since $w_t$ satisfies the Assumption 2), and $w_t \notin \mathcal{F}_t$, then $\sigma w | \mathcal{F}_t$ is sub-Gaussian with parameter $\sigma w$. Additionally, system (1) is linear in unknown parameters $\theta_*$. From Theorem 1, the $\{\mathcal{F}_t\}_{t \geq 1}$-adapted process $\{\phi(z_t)\}_{t \geq 1}$ satisfies the $(1, s_\phi^2 I_{n_\phi}, p_\phi)$-BMSB condition for some $s_\phi > 0$ and $p_\phi \in (0, 1]$.

To complete the proof, it is left to show that for any $\delta \in (0, 1)$, there exists a $\bar{\Gamma} \succ s_\phi^2 I_{n_\phi}$, such that

$$\mathbb{P}\left(\sum_{t=1}^{T} \phi(z_t)\phi^\intercal(z_t) \not\succeq T\bar{\Gamma}\right) \leq \delta.$$

To see this, note that for $\bar{b}_\phi = \sup_{t \geq 0} \mathbb{E}\big[\|\phi(z_t)\|_2^2\big]$, we have:

$$\mathbb{P}\left(\sum_{t=1}^{T} \phi(z_t)\phi^\intercal(z_t) \not\succeq \frac{\bar{b}_\phi T}{\delta} I_{n_\phi}\right) = \mathbb{P}\left(\lambda_{\max}\left(\sum_{t=1}^{T} \phi(z_t)\phi^\intercal(z_t)\right) \succ \lambda_{\max}\left(\frac{\bar{b}_\phi T}{\delta} I_{n_\phi}\right)\right)$$

$$= \mathbb{P}\left(\Big\|\sum_{t=1}^{T} \phi(z_t)\phi^\intercal(z_t)\Big\|_2 \succ \frac{\bar{b}_\phi T}{\delta}\right)$$

$$\leq \frac{\delta \mathbb{E}\Big[\big\|\sum_{t=1}^{T} \phi(z_t)\phi^\intercal(z_t)\big\|_2\Big]}{\bar{b}_\phi T}.$$

In addition, we have:

$$\mathbb{E}\left[\Big\|\sum_{t=1}^{T} \phi(z_t)\phi^\intercal(z_t)\Big\|_2\right] \leq \sum_{t=1}^{T} \mathbb{E}\big[\|\phi(z_t)\phi^\intercal(z_t)\|_2\big] \leq \sum_{t=1}^{T} \mathbb{E}\big[\|\phi(z_t)\|_2^2\big] \leq T \sup_{t \geq 0} \mathbb{E}\big[\|\phi(z_t)\|_2^2\big],$$

which implies that:

$$\mathbb{P}\left(\sum_{t=1}^{T} \phi(z_t)\phi^\intercal(z_t) \not\succeq T\bar{\Gamma}\right) \leq \delta, \tag{7}$$

where $\bar{\Gamma} = \dfrac{\bar{b}_\phi}{\delta} I_{n_\phi}$. since $z_t \in \mathcal{Z}$ for all $t \geq 0$, with $\mathcal{Z}$ being a compact set (due to the system's LISS property and features $\phi(\cdot)$ being real-analytic), such a bounded $\bar{b}_\phi$ exists, completing the proof. $\qquad\square$

## B.2   Proof of Corollary 1

*Proof.* We start the proof by stating the following lemma which is extension of Theorem 1 to the case with $u_t = \pi(x_t) + \eta_t$.

**Lemma 3.** *Let $u_t = \pi(x_t) + \eta_t$, $\pi(\cdot)$ is real-analytic, $\eta_t$ satisfies Assumption 3. Consider the filtration $\mathcal{F}_t = \mathcal{F}(w_0, \cdots, w_{t-1}, x_0, \cdots, x_t, \pi(x_0), \cdots, \pi(x_t), \eta_0, \cdots, \eta_t)$. Suppose Assumptions 1, 2 hold and that the closed-loop system $x_{t+1} = \theta_* \phi(x_t, \pi(x_t) + \eta_t) + w_t$ satisfies Assumption 4 for both $w_t$ and $\eta_t$. Then there exist $s_\phi > 0$, and $p_\phi \in (0, 1)$ such that the $\{\mathcal{F}_t\}_{t \geq 1}$-adapted process $\{\phi(x_t, u_t)\}_{t \geq 1}$ satisfies the $(1, s_\phi^2 I_{n_\phi}, p_\phi)$-BMSB condition.*

*Proof of Lemma 3.* The proof closely follows the steps of Theorem 1. First, observe that $u_t = \pi(x_t) + \eta_t$. Since $\eta_t$ satisfies the conditions outlined in Assumption 3, and the closed-loop system $x_{t+1} = \theta_* \phi(x_t, \pi(x_t) + \eta_t) + w_t$ adheres to Assumption 4 with respect to both $w_t$ and $\eta_t$, there exist functions $\gamma \in \mathcal{K}$ and $\beta \in \mathcal{KL}$ such that for all $t \geq 0$ the following holds:

$$x_t \in \mathcal{X} = \left\{x \in \mathbb{R}^n : \|x\|_2 \leq \beta(\rho_x, 0) + \gamma(\rho)\right\}$$

with parameters $\rho_x \geq \|x_0\|_2$ and $\rho \geq \max(w_{\max}, u_{\max})$. Let $\mathcal{Z} = \mathcal{X} \times \mathcal{U}$, where $\mathcal{U}$ is a compact set containing $u_t$ for all $t \geq 0$. Thus, $z_t \in \mathcal{Z}$ for all $t \geq 0$. The set $\mathcal{Z}$ is a compact subset of $\mathbb{R}^{n_x + n_u}$. The remaining part of the proof, specifically to show that $\sup_{\mathcal{F}_t, \, t \geq 0} \sup_{v \in \mathcal{V}} \mathbb{E}\big[(v^\intercal \phi(z_{t+1}))^4 \mid \mathcal{F}_t\big] < \infty$ follows similarly to the proof in Theorem 1.

It remains to show that $\inf_{\mathcal{F}_t,\,t\geq 0}\ \inf_{v\in\mathcal{V}}\mathbb{E}\big[\big(v^{\mathsf{T}}\phi(z_{t+1})\big)^2\mid\mathcal{F}_t\big] > 0$. This can be shown as follows:

$$
\inf_{\mathcal{F}_t,\,t\geq 0}\ \inf_{v\in\mathcal{V}}\mathbb{E}\left[\big(v^{\mathsf{T}}\phi(z_{t+1})\big)^2\mid\mathcal{F}_t\right] = \inf_{\mathcal{F}_t,\,t\geq 0}\ \inf_{v\in\mathcal{V}}\mathbb{E}\left[\big(v^{\mathsf{T}}\phi(x_{t+1},u_{t+1})\big)^2\mid\mathcal{F}_t\right]
$$

$$
= \inf_{\mathcal{F}_t,\,t\geq 0}\ \inf_{v\in\mathcal{V}}\mathbb{E}\left[\left(v^{\mathsf{T}}\phi\big(x_{t+1},\pi(x_{t+1})+\eta_{t+1}\big)\right)^2\mid\mathcal{F}_t\right]
$$

$$
= \inf_{\mathcal{F}_t,\,t\geq 0}\ \inf_{v\in\mathcal{V}}\mathbb{E}\Bigg[\bigg(v^{\mathsf{T}}\phi\Big(\theta_*\phi(z_t)+w_t,
$$

$$
\pi\big(\theta_*\phi(z_t)+w_t\big)+\eta_{t+1}\Big)\bigg)^2\mid\mathcal{F}_t\Bigg].
$$

Since $z_t\in\mathcal{F}_t$ and $w_t,\eta_{t+1}\notin\mathcal{F}_t$ then $z_t$ is treated as constant while $w_t,\eta_{t+1}$ are considered random variables. Then, by the continuity of $\phi$ we conclude that

$$
\inf_{\mathcal{F}_t,\,t\geq 0}\ \inf_{v\in\mathcal{V}}\mathbb{E}\left[\big(v^{\mathsf{T}}\phi(z_{t+1})\big)^2\mid\mathcal{F}_t\right] = \inf_{z\in\mathcal{Z}}\ \inf_{v\in\mathcal{V}}\mathbb{E}\left[\bigg(v^{\mathsf{T}}\phi\Big(\underbrace{\theta_*\phi(z)+w}_{=:\,h(z,w)},\pi\big(\underbrace{\theta_*\phi(z)+w}_{=:\,h(z,w)}\big)+\eta\Big)\bigg)^2\right]
$$

$$
= \inf_{z\in\mathcal{Z}}\ \inf_{v\in\mathcal{V}}\mathbb{E}\left[\bigg(v^{\mathsf{T}}\phi\Big(h(z,w),\pi\big(h(z,w)\big)+\eta\Big)\bigg)^2\right]
$$

where $w$ and $\eta$ are independent random variables constrained as described in Assumptions 2 and 3. By letting $\mathcal{N}_v^z=\big\{(w,\eta)\in\mathcal{W}\times\mathcal{U}:\ v^{\mathsf{T}}\phi\big(h(z,w),\pi\big(h(z,w)\big)+\eta\big)=0\big\}$, similar to (6) we have:

$$
\inf_{z\in\mathcal{Z}}\ \inf_{v\in\mathcal{V}}\mathbb{E}\left[\bigg(v^{\mathsf{T}}\phi\Big(h(z,w),\pi\big(h(z,w)\big)+\eta\Big)\bigg)^2\right]
$$

$$
= \inf_{z\in\mathcal{Z}}\ \inf_{v\in\mathcal{V}}\mathbb{E}\left[\bigg(v^{\mathsf{T}}\phi\Big(h(z,w),\pi\big(h(z,w)\big)+\eta\Big)\bigg)^2\mid (w,u)\notin\mathcal{N}_v^z\right]
$$

$$
\times\left(1-\sup_{z\in\mathcal{Z}}\ \sup_{v\in\mathcal{V}}\mathbb{P}\Big((w,\eta)\in\mathcal{N}_v^z\Big)\right).
$$

We aim to show that $\lambda^{n_x+n_u}(\mathcal{N}_v^z)=0$ by applying Lemma 2. Note that $\phi(\cdot)$, $h(\cdot)$, and $\pi(\cdot)$ are real-analytic. To use the results of Lemma 2, we need to establish that $v^{\mathsf{T}}\phi\Big(h(z,w),\pi\big(h(z,w)\big)+\eta\Big)$ is non-zero for any $v\in\mathcal{V}$. First, observe that:

$$
v^{\mathsf{T}}\phi\big(h(z,w),\pi(h(z,w))+\eta\big) = \sum_{i=1}^{n_\phi}v_i\phi^i\big(h(z,w),\pi(h(z,w))+\eta\big).
$$

Now consider two scenarios:

- All components of $\pi\big(h(z,w)\big)$ are linearly independent with any component of $h(z,w)$.

- At least one component of $\pi\big(h(z,w)\big)$ is linearly dependent with one or more components of $h(z,w)$.

In both cases, due to the additive nature of $\eta$, all the functions $\phi^i\big(h(z,w),\pi(h(z,w))+\eta\big)$ with $i=1,\cdots,n_\phi$ are linearly independent, ensuring that $v^{\mathsf{T}}\phi\big(h(z,w),\pi(h(z,w))+\eta\big)\not\equiv 0$ for any $v\in\mathcal{V}$. The remainder of the proof follows similarly to the argument in Theorem 1. $\qquad\square$

Using this Lemma, Theorem 4, and reasoning similar to that in the proof of Theorem 2, the proof can be completed. $\qquad\square$

# C  Proofs for Theorem 3, Corollary 2, and Corollary 3

## C.1  Proof of Theorem 3

*Proof.* The proof follows from applying the following meta-theorem on the convergence rate of SME.

**Theorem 5** (SME meta-theorem (Li et al., 2024))**.** *Consider a general time series model with linear responses as follows:*

$$x_t = \theta_* y_t + w_t, \quad t \geq 0.$$

*Also, define the filtration $\mathcal{F}_t = \mathcal{F}(w_0, \cdots, w_{t-1}, y_0, \cdots, y_t)$. Assume the following conditions are met:*

- *$w_t$ are i.i.d. with variance $\sigma_w^2 I_{n_x}$, and box-constrained, i.e., $w_t \in \mathcal{W} = \{w \in \mathbb{R}^{n_x} : \|w\|_\infty \leq w_{\max}\}$ for some $w_{\max} > 0$.*

- *$\{y_t\}_{t\geq 1}$ is an $\{\mathcal{F}_t\}_{t\geq 1}$-adapted random process satisfying the $(k, s_y^2 I_{n_y}, p_y)$-BMSB condition.*

- *There exists $b_y > 0$ such that $\|y_t\|_2 \leq b_y$ almost surely for all $t \geq 0$.*

- *For any $\ell > 0$, there exists $q_w(\ell) > 0$, such that for any $1 \leq j \leq n$ and $t \geq 0$, we have $\mathbb{P}(w_t^j + w_{\max} \leq \ell) \geq q_w(\ell) > 0$, $\mathbb{P}(w_{\max} - w_t^j \leq \ell) \geq q_w(\ell) > 0$.*

*Then for any $m \geq 1$ and any $\delta \in (0,1)$, when $T > m$, the diameter of the uncertainty set*

$$\Theta_T = \bigcap_{t=0}^{T-1} \left\{ \hat{\theta} : x_t - \hat{\theta} y_t \in \mathcal{W} \right\}$$

*satisfies:*

$$
\mathbb{P}\left(\operatorname{diam}(\Theta_T) > \delta\right) \leq 544 \frac{T}{m} n_y^{2.5} \log(a_2 n_y) a_2^{n_y} \exp(-a_3 m)
$$
$$
+ 544 n_x^{2.5} n_y^{2.5} \log(a_4 n_x n_y) a_4^{n_x n_y} \left(1 - q_w\left(\frac{a_1 \delta}{4\sqrt{n_x}}\right)\right)^{\lceil T/m \rceil},
$$

*where $a_1 = \frac{s_y p_y}{4}$, $a_2 = \frac{64 b_y^2}{s_y^2 p_y^2}$, $a_3 = \frac{p_y^2}{8}$, $a_4 = \max\left(\frac{4 b_y \sqrt{n_x}}{a_1}, 1\right)$.*

Observe that system (1) is linear in the unknown parameters $\theta_*$, and we can prove Theorem 3 by showing that the $\{\mathcal{F}_t\}_{t\geq 1}$-adapted process $\{\phi(z_t)\}_{t\geq 1}$ and $w_t$ meet the conditions of the meta-theorem. By Assumptions 2 and 5, and since $w_t \notin \mathcal{F}_t$, the noise $w_t$ fulfills all the requirements of the meta-theorem. Moreover, according to Theorem 1, the $\{\mathcal{F}_t\}_{t\geq 1}$-adapted process $\phi(z_t)_{t\geq 1}$ satisfies the $(1, s_\phi^2 I_{n_\phi}, p_\phi)$-BMSB condition for some $s_\phi > 0$ and $p_\phi \in (0,1]$. Lastly, since the system is LISS, we have $z_t \in \mathcal{Z}$ for all $t \geq 0$, where $\mathcal{Z}$ is the compact set defined in the proof of Theorem 1. Therefore, there exists a constant $b_\phi > 0$ such that $\sup_{t\geq 0} \|\phi(z_t)\|_2 \leq b_\phi$, completing the proof of the theorem.

Explicitly, this means that for any $m \geq 1$, for any $\delta \in (0,1)$, when $T > m$, we have:

$$
\mathbb{P}\left(\operatorname{diam}(\Theta_T) > \delta\right) \leq 544 \frac{T}{m} n_\phi^{2.5} \log(a_2 n_\phi) a_2^{n_\phi} \exp(-a_3 m)
$$
$$
+ 544 n_x^{2.5} n_\phi^{2.5} \log(a_4 n_x n_\phi) a_4^{n_x n_\phi} \left(1 - q_w\left(\frac{a_1 \delta}{4\sqrt{n_x}}\right)\right)^{\lceil T/m \rceil}, \tag{8}
$$

where $a_1 = \frac{s_\phi p_\phi}{4}$, $a_2 = \frac{64 b_\phi^2}{s_\phi^2 p_\phi^2}$, $a_3 = \frac{p_\phi^2}{8}$, $a_4 = \frac{16 b_\phi \sqrt{n_x}}{s_\phi p_\phi}$. □

## C.2 Proof of Corollary 2

Let us provide two example distributions, truncated Gaussian and uniform, along with their corresponding $q_w(\cdot)$ (from (Li et al., 2024)):

- If $w_t$ follows a uniform distribution on $[-w_{\max}, w_{\max}]^{n_x}$, then $q_w(\ell) = c_w \ell$ with $c_w = \frac{1}{2w_{\max}}$.
- If $w_t$ follows a truncated-Gaussian distribution on $[-w_{\max}, w_{\max}]^{n_x}$, generated by a Gaussian distribution with zero mean and covariance matrix $\sigma_w^2 I_{n_x}$, then $q_w(\ell) = c_w \ell$ with $c_w = \frac{1}{\min(\sqrt{2\pi}\sigma_w, 2w_{\max})} \exp(\frac{-w_{\max}^2}{2\sigma_w^2})$.

Now, fix $\epsilon \in (0, 1)$. We want to show that if $q\left(\frac{a_1 \delta}{4\sqrt{n_\phi}}\right) = c_w \frac{a_1 \delta}{4\sqrt{n_\phi}}$ and we choose $m$ such that

$$m \geq \frac{1}{a_3}\left(\log\left(\frac{T}{\epsilon}\right) + n_\phi \log(a_2) + 2.5\log(n_\phi) + \log\log(a_2 n_\phi) + \log(544)\right), \tag{9}$$

then for all $T \geq m$, we have

$$\delta \leq O\left(\frac{m\sqrt{n_x}\log\left(\frac{1}{\epsilon}\right) + mn_x^{1.5}n_\phi \log\left(\frac{b_\phi\sqrt{n_x}}{s_\phi p_\phi}\right)}{c_w s_\phi p_\phi T}\right) \tag{10}$$

with probability at least $1 - 2\epsilon$.

Let the two terms in the right-hand side of (8) be denoted by "term 1" and "term 2". We proceed with the proof in two steps as follows:

**Step 1: showing that with the choice of $m$ in (9), term 1 $\leq \epsilon$ :**

With this choice of $m$, it is straightforward to see that

$$544 T n_\phi^{2.5} \log(a_2 n_\phi) a_2^{n_\phi} \exp(-a_3 m) \leq \epsilon,$$

and thus term 1 $\leq \epsilon$.

**Step 2: letting term 2 $= \epsilon$ and showing that $\delta$ satisfies the inequality in (10):**

Assuming without loss of generality that $\frac{T}{m}$ is an integer, note that term 2 $= \epsilon$ implies:

$$q_w\left(\frac{a_1 \delta}{4\sqrt{n_x}}\right) = \left(1 - \left(\frac{\epsilon}{544 n_x^{2.5} n_\phi^{2.5} \log(a_4 n_x n_\phi) a_4^{n_x n_\phi}}\right)^{\frac{m}{T}}\right)$$

$$\leq -\log\left(\frac{\epsilon}{544 n_x^{2.5} n_\phi^{2.5} \log(a_4 n_x n_\phi) a_4^{n_x n_\phi}}\right)^{\frac{m}{T}}$$

$$= -\frac{m}{T}\log\left(\frac{\epsilon}{544 n_x^{2.5} n_\phi^{2.5} \log(a_4 n_x n_\phi) a_4^{n_x n_\phi}}\right)$$

$$= \frac{m}{T}\left(\log\left(\frac{1}{\epsilon}\right) + \log(a_4) n_x n_\phi + 2.5\log(n_x n_\phi) + \log\log(a_4 n_x n_\phi) + \log(544)\right).$$

If $q_w\left(\frac{a_1 \delta}{4\sqrt{n_x}}\right) = c_w \frac{a_1 \delta}{4\sqrt{n_x}}$ for some constant $c_w > 0$, then:

$$\delta \leq \frac{4\sqrt{n_x} m}{c_w a_1 T}\left(\log\left(\frac{1}{\epsilon}\right) + \log(a_4) n_x n_\phi + 2.5\log(n_x n_\phi) + \log\log(a_4 n_x n_\phi) + \log(544)\right)$$

$$\leq \frac{16\sqrt{n_x} m}{c_w s_\phi p_\phi T} O\left(\log\left(\frac{1}{\epsilon}\right) + n_x n_\phi \log\left(\frac{16 b_\phi \sqrt{n_x}}{s_\phi p_\phi}\right)\right)$$

$$\leq O\left(\frac{m\sqrt{n_x}\log\left(\frac{1}{\epsilon}\right) + mn_x^{1.5}n_\phi \log\left(\frac{b_\phi n_x}{s_\phi p_\phi}\right)}{c_w s_\phi p_\phi T}\right).$$

Combining these two steps, we conclude that, with probability at least $1 - 2\epsilon$,

$$\text{diam}(\Theta_T) \leq O\left(\frac{m\sqrt{n_x}\log\left(\frac{1}{\epsilon}\right) + mn_x^{1.5}n_\phi \log\left(\frac{b_\phi n_x}{s_\phi p_\phi}\right)}{c_w s_\phi p_\phi T}\right).$$

### C.3 Proof of Corollary 3

This corollary's proof builds on Lemma 3 in Appendix B.2 and closely aligns with the proofs of Theorem 3 and Corollary 2.

## D   Numerical Experiments

This section provides details on the simulation experiments.

### D.1   Pendulum

The ground truth for the unknown parameters for pendulum example in Example 1 is set to be

$$m = 0.1 \text{ (kg)}, \quad l = 0.5 \text{ (m)},$$

and discretization time step in our numerical experiments is $dt = 0.01$ (s). The control input is a simple feedback controller $u_t = -k\dot{\alpha}_t + \eta_t$. In Figures 1a and 1b we choose $k = 2$ and in Figures 2a, 2b and 3 we choose $k = 0.1$. Note there are two unknown parameters in this pendulum example as follows:

$$\theta_1 = \frac{1}{l}, \quad \theta_2 = \frac{1}{ml^2}.$$

### D.2   Quadrotor

The ground truth for the unknown parameters for quadrotor example in Example 2 is set to be

$$m = 0.468 \text{ (kg)},$$

$$I_{xx} = 4.856 \times 10^{-3} \text{ (kg/m}^2), \quad I_{yy} = 4.856 \times 10^{-3} \text{ (kg/m}^2), \quad I_{zz} = 8.801 \times 10^{-3} \text{ (kg/m}^2).$$

The discretization time step in our numerical experiments is $dt = 0.01$ (s). The control input is a control policy plus i.i.d. noise. The control policy on altitude and the three Euler angles is borrowed from (Alaimo et al., 2013). The controller gains in our numerical experiments are chosen as:

$$kp_z = 0.75, \quad kd_z = 1.25,$$
$$kp_\phi = 0.03, \quad kd_\phi = 0.00875,$$
$$kp_\theta = 0.03, \quad kd_\theta = 0.00875,$$
$$kp_\psi = 0.03, \quad kd_\psi = 0.00875.$$

Note that there are seven unknown parameters in this quadrotor example, as follows:

$$\theta_1 = \frac{1}{m},$$

$$\theta_2 = \frac{1}{I_{xx}}, \quad \theta_3 = \frac{I_{yy} - I_{zz}}{I_{xx}}, \quad \theta_4 = \frac{1}{I_{yy}}, \quad \theta_5 = \frac{I_{zz} - I_{xx}}{I_{yy}}, \quad \theta_6 = \frac{1}{I_{zz}}, \quad \theta_7 = \frac{I_{xx} - I_{zz}}{I_{zz}}.$$

Figure 4 displays the uncertainty set estimated by SME for the seven unknown parameters in the quadrotor example for various trajectory lengths, with $\eta_t$ and $w_t$ being i.i.d. samples from truncated-Gaussian distributions. The uncertainty sets are observed to shrink as the trajectory length increases, consistent with our theoretical results. Note that the ground truth value is contained within all the uncertainty sets.

## E   Numerical Estimation of BMSB Parameters $(s_\phi, p_\phi)$

We compare the empirical rates of LSE and SME with their theoretical counterparts in Section 4. The theoretical results presented in Theorem 2 and Corollary 2 rely on the parameters $b_\phi$, $\bar{b}_\phi$, and the BMSB parameters $(s_\phi, p_\phi)$. However, the explicit relationship of these parameters with system, noise, and disturbance characteristics such as $n_x$, $n_u$, $n_\phi$, $\sigma_u$, and $\sigma_w$ is not known and we will address this in our future work. Consequently, we estimate these parameters numerically and utilize these

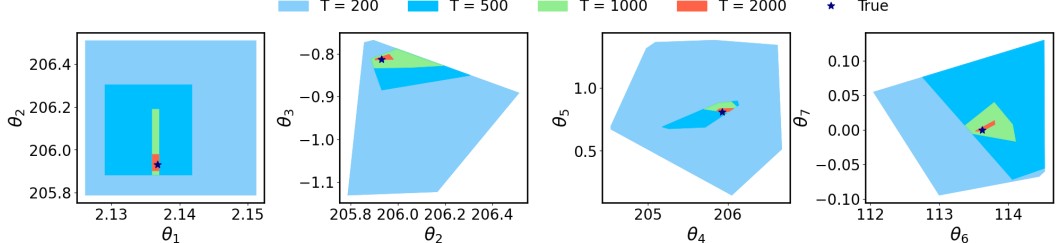

Figure 4: 2D projections of the uncertainty set estimated by SME for the unknown parameters of the quadrotor example. The noises and disturbances are i.i.d generated from `truncated-Gaussian`$(0, 0.5, [-1, 1])$.

estimates to calculate the theoretical rates discussed in Section 4. While $b_\phi$ and $\bar{b}_\phi$ are straightforward to estimate, special attention is required to estimate the BMSB parameters. This section is dedicated to describing this estimation process.

For this, consider a system of the form (1). For this system, our goal is to estimate $s_\phi$ and $p_\phi$, where

$$p_\phi = \inf_{\mathcal{F}_t, t \geq 0} \inf_{\|v\|_2 = 1} \mathbb{P}\left( |v^T \phi(z_{t+1})| \geq s_\phi \mid \mathcal{F}_t \right)$$

numerically. First, observe that $\phi(z_{t+1}) \mid \mathcal{F}_t = \phi(\theta_*^T \phi(z_t) + w_t, u_{t+1})$, where $z_t \in \mathcal{F}_t$. This implies that $\phi(z_{t+1}) \mid \mathcal{F}_t$ is a random variable influenced by $w_t$ and $u_{t+1}$. We proceed by fixing $s_\phi = \bar{s}$ (for some $\bar{s} > 0$) and empirically estimate $p_\phi$. To accomplish this, we first select a time horizon $T$ and generate several trajectories of length $T$ for the system. Let $\mathcal{D}^T$ represent the set of these trajectories, while $\mathcal{D}^t$ denotes a subset containing all trajectories up to $t \leq T$. Additionally, we create multiple vectors $v \in \mathbb{R}^{n_\phi}$ such that $\|v\|_2 = 1$; we refer to this collection as $\bar{\mathcal{V}}$. These vectors are randomly sampled from a Gaussian distribution and subsequently normalized.

We then estimate $\bar{p}$ as:

$$\bar{p} = \min_{t \in [T]} \min_{z \in \mathcal{D}^t} \min_{v \in \bar{\mathcal{V}}} \mathbb{P}\left( |v^T \phi(\theta_*^T \phi(z) + w_t, u_{t+1})| \geq \bar{s} \right).$$

As $T$ increases, along with the number of trajectories and vectors $v$, the minimum estimates will more accurately reflect the infimums. In this context, $\bar{p}$ represents the minimum of $\mathbb{P}\left( |v^T \phi(\theta_*^T \phi(z) + w_t, u_{t+1})| \geq \bar{s} \right)$ across all combinations in $[T] \times \mathcal{D}^T \times \bar{\mathcal{V}}$. For each combination in this set, we estimate the probability $\mathbb{P}\left( |v^T \phi(\theta_*^T \phi(z) + w_t, u_{t+1})| \geq \bar{s} \right)$ using Monte Carlo simulations. This process entails generating multiple random samples based on the distributions of $w_t$ and $u_{t+1}$, verifying whether each sample satisfies the condition $|\bar{v}^T \phi(z)| \geq \bar{s}$, and tallying how many samples meet this criterion. We repeat this procedure for various values of $\bar{s}$ until we identify a pair of $(\bar{s}, \bar{p})$ such that $\bar{p} \in (0, 1)$. The estimated probability is calculated as the ratio of the count of successful samples to the total number of samples. According to the law of large numbers, this ratio converges to the true probability as the number of samples increases. For our estimations, we select $T = 50$, $|\bar{\mathcal{V}}| = 1000$, and $|\mathcal{D}^T| = 20$.

