# OpenReview forum: "Identification of Analytic Nonlinear Dynamical Systems with Non-asymptotic Guarantees"
_NeurIPS.cc/2024/Conference — NeurIPS 2024 poster_

### Official Review · Reviewer_QHWP · 2024-07-08

**Soundness:** 3
**Presentation:** 3
**Contribution:** 2
**Rating:** 5
**Confidence:** 4

**Summary:**

This paper studies the problem of identifying an observable stochastic nonlinear dynamical system, in case the transition function is linearly parametrized and the noise is additive. The authors assume that the feature functions are analytic, and both the inputs and the noises are i.i.d., bounded, semi-continuous and have nonvanishing variances in each coordinate direction. It is also assumed that the system is locally input-to-state stable. Two kinds of estimation methods are studied: the classical least-squares estimate (LSE), which provides point estimates, and a set membership method, which provides region estimates. Finite sample bounds for the performance of LSE are proved, based on a block-martingale-small-ball condition, for both the open-loop and the close-loop cases. The sample complexity of the set membership method is studied under an additional assumption (tight bound on disturbance) and only for open-loop (i.i.d.) inputs. Finally, the authors present some numerical evaluations of LSE and set membership methods on pendulum and quadrotor examples to empirically illustrate their convergence rates.

**Strengths:**

- The presentation is clear, the paper is well-structured, the need assumptions are precisely stated.
- The LSE are widely used, and the set membership approach is also reasonable and practically relevant.
- Sample complexity analysis of system identification methods for nonlinear systems is an important problem. On the other hand, it is more relevant for control theory than to machine learning.
- Finite sample error bounds are provided for both methods under the assumption that the features are analytic. This approach seems original and could be interesting for the community.
- The examples illustrate well the theoretical viability of the analytic features assumption.

**Weaknesses:**

- Many of the assumptions are restrictive, such as bounded noises and inputs. Moreover, the inputs should also be i.i.d. (or should have an additive i.i.d. exploration noise, for the closed-loop case) which is unrealistic.
- The error bounds contain terms which are unknown in practice, such as s_\phi.
- Corollary 1: the controller for the closed-loop case also contains an additive noise term that satisfies the assumptions for the open-loop inputs, and the system with the controller satisfies the stability assumption, in which case the statement becomes a simple consequence of Theorem 1. A case without additional noises on the inputs would have been much more interesting.
- The figures do not show the actual bounds deducted by the paper, they only illustrate the empirical performance of LSE and set membership identification, which is a bit pointless, as they are classical, well-studied methods. I understand that the bounds of the paper are theoretical in nature, so they are conservative and mostly just give the convergence rate, still showing them (for example, on a logarithmic scale) would have been informative.
- The title could be misleading, as the word "analytic" should refer to the feature vectors, but in the current title the term "analytic system identification" could also be understood in a way that the obtained solution is analytic.

**Questions:**

- What is the intuitive meaning of Assumption 5 (tight bound on disturbance)? This should be explained in the paper.

**Limitations:**

There is a section dedicated to the limitations of the work, which is a good thing. On the other hand, this section did not mention some key limitations, such as bounded noises, bounded and i.i.d. inputs, as well was fully observable states.

---

> ### Author Rebuttal · Authors · 2024-08-07
>
> Thanks for your valuable comments!
>
> ---
>
> `Q1`: **Bounded noises/inputs**
>
> `A`: First, we politely point out that, though **linear** control usually considers unbounded noises/inputs, bounded noises/inputs are commonly studied in many **nonlinear** control literature (Mania et al. 2022, Shi et al. 2021, Kim et al. 2024).
>
> Second, even for nonlinear control papers with unbounded noises/inputs, they usually impose **stronger assumptions on system dynamics**, e.g. globally exponentially stable systems (Sattar & Oymak 2022), or global Lipschitz smooth dynamics (Lee et al. 2024). However, in practice, most nonlinear systems can only be locally stable (Slotine & Li 1991). Further, global Lipschitz smoothness doesn't include polynomial systems with order >=3, which have many applications (see Example 2 in our paper, Slotine & Li 1991).
>
> Thus, there is a **tradeoff** between noise assumptions vs. dynamics assumptions. For physical system applications, we think bounded noises/inputs are more practical than globally stable or globally smooth dynamics. After all, most noises in physical systems are bounded, e.g. wind gusts for quadrotors, renewable energy generation for power systems, etc.; and most inputs are also bounded, e.g. bounded thrusts from propellers and bounded energy generation by conventional power plants.
>
> Yet, it is interesting to study convergence rates under unbounded noises but stronger dynamics assumptions. After a quick check, our results still hold under the assumptions in (Sattar & Oymak 2022), but assumptions in other papers need more work.
>
> &nbsp;
>
> ---
>
> `Q1 cont'd & Q3`:  **I.i.d. exploration noises & Cor. 1**
>
> `A`: First, we politely mention that i.i.d. exploration noises are commonly added to existing controllers to provide necessary exploration when learning linear (Simchowitz & Foster 2020, Dean et al. 2019) and nonlinear systems (Sattar et al. 2023, Li et al. 2023). Compared with other exploration methods, e.g. optimism-in-the-face-of-uncertainty (OFU), Thompson sampling (TS), and PE-based recursive constrained optimization (RCO) (Lu & Cannon 2023), exploration noises are popular for the following reasons:
> - Generating i.i.d. noises is usually much **simpler and less computationally demanding** than other methods, e.g. OFU can be intractable for nonlinear control (see Kakade et al. 2020), TS's posterior sampling and RCO are also time-consuming.
> - Exploration noises are a **generic plug-in** to most existing controllers and don't need new control designs as in OFU, etc.
> - **Optimal performance guarantees** can be achieved by additive i.i.d. exploration noises on linear systems (Simchowitz & Foster 2020), which motivates its applications on nonlinear systems.
>
> Regarding the practical challenges and how to address them:
> - Locally stable systems may become **unstable** with large exploration noises. To address this, one can 1) start from a small exploration noise then gradually increase it, 2) adopt a stability certificate and switch to a noiseless stabilizing controller when the system fails the certificate (Fisac et al. 2018).
> - In some applications, the **fluctuations** caused by i.i.d. noises are not desirable. To address this, it is common to replace the i.i.d. noises with sinusoidal noises, which enjoy similar empirical performances (Nesic et al. 2012).
>
> As for Cor. 1 without i.i.d. noises, we note that if the noiseless inputs satisfy BMSB, then our convergence rates still hold. The major difficulty is designing such noiseless inputs and formally proving BMSB.
>
> We will add the discussions to the revised paper. We are happy to discuss more if you have remaining concerns!
>
> &nbsp;
>
> ---
>
> `Q2`: **Unknown $s_\phi$**
>
> `A`: The empirical values of $(s_\phi,p_\phi)$ can be estimated by Monte Carlo simulation (see Fig. 5 in rebuttal.pdf). By definition, $(s_\phi,p_\phi)$ can take multiple values for the same system so we plot them as a curve. When computing the theoretical upper bounds in Thm. 2 (Fig. 2 in rebuttal.pdf), we choose the largest $s_\phi p_\phi$ for better bounds.
>
> The explicit formulas of $(s_\phi, p_\phi)$ call for stronger assumptions on dynamics, and we don't think a generic formula exists for all analytic nonlinear systems. During our research, we obtained formulas of $(s_\phi,p_\phi)$ for quadratic systems by algebraic manipulation, but we didn't use this proof because it cannot be generalized to other systems. It is left as future work to provide explicit forms of $(s_\phi,p_\phi)$ for other systems.
>
> &nbsp;
>
> ---
>
> `Q4`: **Plot theoretical bounds**
>
> `A`: We plot theoretical bounds and empirical performance in Fig. 2 of rebuttal.pdf, both of which have slope -1/2 in a log-log scale, being consistent with our convergence rate.
>
> `Q5 & Q7`: **Title & limitations**
>
> `A`: Thanks for the suggestions! We will revise the title and discuss these limitations.
>
> `Q6`: **Intuition of Ass. 5**
>
> `A`: We will add the following intuitions. Consider the upper bound of a 1-dim $w_t$ bounded by $-w_{\max}\leq w_t\leq w_{\max}$. Ass. 5 assumes a tight bound, meaning that there is a non-vanishing probability for $w_t$ to visit around $w_{\max}$, i.e. $P(w_{\max}-\epsilon\leq w_t\leq w_{\max})>0$ for any $\epsilon>0$.
>
> &nbsp;
>
> ---
>
> References
>
> Kakade et al. 2020: Information-theoretic regret bounds for online nonlinear control, Neurips.
>
> Shi et al. 2021: Meta-adaptive nonlinear control: Theory and algorithms, Neurips.
>
> Lee et al. 2024: Active Learning for Control-Oriented Identification of Nonlinear Systems.
>
> Kim & Lavaei 2024: Online Bandit Control with Dynamic Batch Length and Adaptive Learning Rate.
>
> Slotine & Li 1991: Applied nonlinear control
>
> Lu & Cannon 2023: Robust adaptive model predictive control with persistent excitation conditions, Automatica.
>
> Fisac et al. 2018: A general safety framework for learning-based control in uncertain robotic systems, IEEE-TAC.
>
> Nesic et al. 2012, A framework for extremum seeking control of systems with parameter uncertainties. IEEE-TAC.

---

> > ### Comment · Reviewer_QHWP · 2024-08-09
> >
> > Thank you for your comprehensive answers and for your additional experiments. I can accept most of your arguments and even though I still think that the boundedness assumption is a weakness of the paper and having typically unknown quantities in the bounds is also an issue, I have raised my rating based on the other points.

---

> > > ### Author Response · Authors · 2024-08-13
> > >
> > > Thank you very much for reading our responses and raising your score! We are glad to hear that you can accept most of our arguments! Regarding the bounded noises assumption, we will incorporate the discussions above into our revised manuscript, and for future work, we will continue exploring the convergence rate under weaker assumptions on noises yet stronger assumptions on system dynamics as considered in the literature mentioned in our responses. Regarding the unknown $s_\phi, p_\phi$, we will add the numerical estimation methods and the plots of these values to our revised manuscript, as well as the explicit formulas of these variables in special cases as illustrating examples. We hope these revisions can alleviate your concerns! Thank you again for your positive feedback!

---

### Official Review · Reviewer_wxwx · 2024-07-13

**Soundness:** 3
**Presentation:** 3
**Contribution:** 3
**Rating:** 6
**Confidence:** 3

**Summary:**

The manuscript provides theoretical guarantees for nonlinear system identification using non-active i.i.d. noises, extending from linear systems to linearly parametrized nonlinear systems with analytic feature functions.  The findings demonstrate that non-active i.i.d. noises are capable of efficiently learning these systems with a non-asymptotic convergence rate.

**Strengths:**

Training data selection is a crucial factor in ensuring the generalizability and robustness of identification algorithms. This is particularly important to linearly parametrized nonlinear systems, as there is still a theoretical gap regarding the effectiveness of non-active i.i.d. noise exploration. The study is well-motivated and has great potential for practical application. The authors establish conditions on noise that guarantee probabilistic persistent excitation for nonlinear dynamical systems, as defined by the BMSB condition. The findings in this study provide theoretical support for designing training data and identifying nonlinear systems.

**Weaknesses:**

Although the authors present an interesting idea, the manuscript could be improved by addressing the following concerns: 1. It is suggested to discuss the potential limitations of the non-asymptotic convergence rate in practical applications. 2. The analysis of the numerical experiments lacks depth. Could the authors further explore the convergence characteristics and the advantages and disadvantages of using i.i.d. noise for excitation? 3. The proof sketches could benefit from additional clarifications to enhance readability. For example, in line 234, the meaning of $\delta$ and $\bar{b}_\phi$ could be re-mentioned.

**Questions:**

Can the authors please clarify the implications of the assumptions regarding bounded noises in more practical scenarios? It would be highly beneficial to delve deeper into the impact of these theoretical assumptions on real-world applications.

**Limitations:**

It is recommended to further explore the potential limitations of this  theoretical guarantees in practical applications. For instance, whether non-asymptotic convergence implies reduced data collection efficiency for some high-dimensional systems, and whether it is still preferable to actively design experiments in certain scenarios.

---

> ### Author Rebuttal · Authors · 2024-08-06
>
> Thank you very much for your helpful suggestions! We discuss your concerns and suggestions below. (For figures, please refer to rebuttal.pdf)
>
> -----
>
> `Weakness 1 & Limitation`:
>
> **On actively designing experiments in certain scenarios**
>
> `A:` In certain scenarios, active exploration can be preferable despite the theoretical guarantees of non-active exploration. For instance, consider the following scenarios:
> 1. Some feature functions are sharper in certain regions away from the origin. For example, in a system described by $\dot x = \theta_{*} u^{3}+w_t$, $u^3$ is flat around $u=0$ but sharp around $u=3$. Active exploration can collect more useful data in these sharp regions, improving convergence performance over non-active exploration. Figure 4 in rebuttal.pdf shows such performance;
>
> 2. For a stable system, inactive exploration only explores around an equilibrium point. But in some cases, the exploration can be boosted by deliberately driving the system to other larger states because larger states can increase the signal-to-noise ratio and improve the information in the data collected;
>
> 3. For systems that are very sensitive to noises, the system may become unstable or even suffer cascading failures after adding exploration noises. In those systems, it is better to use a carefully designed controller that can actively explore while being safe.
>
> &nbsp;
>
> **On data efficiency in high-order systems**
>
> `A:` Data efficiency decreases as the system dimension increases. This is evident in simulations of a cascade system with quadratic sub-systems as shown in Figure 3 in rebuttal.pdf.
>
> &nbsp;
>
> **Other potential limitations of theoretical guarantees in practical applications**
>
> `A:`  Real-world disturbances are more complex than the i.i.d. noises considered in this paper. Disturbances can be influenced by factors like fluid and aerodynamic dynamics, which may not be captured accurately and can include non-analytic components. Modeling may also miss high-frequency errors. In these scenarios, our convergence rate may become invalid. It is an exciting direction to test LSE in realistic scenarios and see when the convergence rate holds and conduct new convergence analysis when existing bounds fail.
>
> &nbsp;
>
> ----
>
>
>
> `Weakness 2`
>
> **On convergence characteristics**
>
> `A:` We provide additional simulation results to explore the following convergence characteristics in rebuttal.pdf.
>
> 1. State Dimensions-- Increasing the dimension leads to worse convergence performance, consistent with theoretical bounds (see Figure 3);
>
> 2. Exploration Noise-- In a pendulum example, higher exploration noises ($\sigma_{u}$) improve convergence performance (Figure 1.a), which is consistent with our theoretical bound as shown by a decrease in $1/s_{\phi}p_{\phi}$ with increased $\sigma_{u}$ (Figure 1.b);
>
> 3. Disturbance Noise-- The effect of disturbance noise ($\sigma_{w}$) on convergence varies by system. In a pendulum example, higher $\sigma_{w}$ worsens convergence (Figure 1.c), which is consistent with terms capturing its effect in our theoretical bound ($\sigma_{w}/s_{\phi}p_{\phi}$ in Figure 1.d).
>
> &nbsp;
>
> **Pros and Cons of i.i.d. exploration noises**
>
> `A:` The major advantage is the simplicity and generality of this approach. One does not need an additional special design for different systems but can simply add i.i.d. noises to the existing controller.  Besides, it saves computation time to compute active exploration design. Another advantage is that i.i.d. exploration makes it easy to balance exploration and exploitation: this can be done by controlling the size of the exploration noises vs. the near-optimal nominal controller.
>
> Most disadvantages of i.i.d. exploration noises are discussed in our response to `Weakness 1 & Limitation` under **on actively designing experiments in certain scenarios**. In addition, i.i.d. exploration noises may create fluctuations that are undesirable for some applications. It is usually recommended to use sinusoidal noises instead for more steady inputs.
>
>
> &nbsp;
>
> ----
>
>
>
>
> `Weakness 3:` **Readability of the Proof Sketch**
>
> `A:` In the revised manuscript, we will re-mention the meanings of these important variables and concepts. We will also add more details to the proof sketch to improve the readability.
>
> &nbsp;
>
> ----
>
>
>
> `Question:` **Practical impacts of bounded Noise Assumption**
>
> `A:` Firstly, most noises in practical scenarios are arguably bounded. For example, the forces/thrusts by wind gusts or fluid dynamics are bounded for robots/mechanics in the air or in the water, the power generations are also bounded for a time step, which is usually a small time period, etc. However, the bounds of the noises are usually unknown. In many cases, only a conservative upper bound can be acquired. Though LSE does not require the knowledge of the noise bounds during implementation, our convergence rate bound can become over-conservative if the noise bounds are also conservative. Another potential issue is when the upper bounds are not conservative but fail to consider a few outliers. In this case, the convergence rate may still hold because the constants in the convergence rate are usually quite conservative. However, the major challenge comes from stability: the few outliers may drive the system to be unstable, thus causing unsafe system behaviors or even causing numeric issues during system identification.
>
> ----
>
> We will add the discussions above to the revised manuscript. We are happy to discuss more if the reviewer has more questions.

---

> > ### Comment · Reviewer_wxwx · 2024-08-11
> > **Official Comment by Reviewer wxwx**
> >
> > Thank you for the detailed responses and the additional simulation. The responses have addressed most of my concerns. I believe my initial rating still reflects my overall assessment, so I will leave it as is.

---

> > > ### Author Response · Authors · 2024-08-13
> > >
> > > Thank you very much for reading our responses and your positive feedback! We are glad that we have addressed most of your concerns!

---

### Official Review · Reviewer_on2R · 2024-07-15

**Soundness:** 3
**Presentation:** 2
**Contribution:** 3
**Rating:** 5
**Confidence:** 3

**Summary:**

The authors study the problem of system identification from a trajectory generated by an unknown linearly parametrized nonlinear system whose nonlinearity is an analytic function. Specifically, they theoretically analyze to estimators: the least squares estimator, and the set membership estimator. Both estimators, while being widely used in practice, do not have theoretical guarantees in many of the settings in which they are applied. The authors give such guarantees, and also conduct numerical experiments on certain nonlinear systems.

**Strengths:**

The authors provide non asymptotic theoretical guarantees for common system identification estimators in settings more broad than those covered by existing theoretical results. As far as I can tell the results are new and the argumentation is sound. They also conduct experiments which verify that such estimators converge to the true system parameters in practice.

**Weaknesses:**

While the theoretical arguments presented seem sound, I am not sure that the extension of system identification results to the case of linearly parametrized smooth nonlinear systems is sufficiently significant from a theoreitcal perspective.

**Questions:**

If instead of assuming that the nonlinearity is analytic, one assumed directly that there does not exist an open set on which it vanishes identically, would the proof carry through?

**Limitations:**

The authors adequately adress the limitations of their work.

---

> ### Author Rebuttal · Authors · 2024-08-06
>
> Thank you very much for your constructive comments! We will address your concerns below:
>
> ---
>
>
> `Weakness:`
> >While the theoretical arguments presented seem sound, I am not sure that the extension of system identification results to the case of linearly parametrized smooth nonlinear systems is sufficiently significant from a theoretical perspective.
>
> `Answer:` Thanks for raising your concern! Though seemingly straightforward, the extension from linear systems to linearly parameterized nonlinear systems is actually **highly nontrivial as reflected by the development of learning-based control literature in the past few years**. In around 2020, the learning-based control community also thought this extension could be straightforward after extensive research on learning-based *linear* control. Unfortunately, the analysis of linearly parameterized nonlinear systems turned out to be much more challenging. For example, the efforts on straightforward extension from linear systems got stuck at bilinear systems (Sattar et al. 2022), which leaves the study of more general linearly parameterized nonlinear systems as an open question. Later, (Mania et al. 2022) provide a quite surprising counter-example, which shows that direct extension of linear control learning approaches may fail in some linearly parameterized nonlinear systems. This counter-example motivates new designs for nonlinear systems (Kowshik et al. 2021, Khosravi et al. 2023) instead of following the footsteps of learning-based linear control.
>
> However, our results indicate that a **much larger class** of linearly parameterized nonlinear systems exists than bilinear systems that still enjoy similar performance as linear systems. Our results greatly **reduce the gaps** in our understanding of linearly parameterized nonlinear systems compared with linear systems. In addition, our results actually support the reviewer's and the community's initial intuition: linear systems and linearly parameterized nonlinear systems indeed share a lot of similarities, as long as nonlinear systems are analytic. This is an interesting and novel message to the learning-based control community.
>
> Theoretically, the major challenge of this work is to identify the largest possible yet reasonable class of systems that enjoys similar performance to linear systems. At first, we established this result for quadratic systems via pretty involved algebraic manipulations, but later, we realized that by leveraging the Paley-Zygmund concentration inequality (Petrov 2007) and the properties of analytic functions, we can generalize the results to all analytic nonlinear feature functions.
>
> The significance of this work is also recognized by Reviewer Dfmt, who finds our work "greatly extends the original results to a much larger class of systems", and by Reviewer wxwx, who thinks this work "has great potential for practical application."
>
> We are happy to provide more discussions if you have remaining questions or concerns! We will add the explanations above to the revised paper.
>
> &nbsp;
>
> ----
>
> `Question`:
> >If instead of assuming that the nonlinearity is analytic, one assumed directly that there does not exist an open set on which it vanishes identically, would the proof carry through?
>
> `Answer:` Thanks for raising this question! If the system is 1-dimensional, then it is enough to assume directly that there does not exist a non-empty open set on which the feature function vanishes identically. However, for multi-dimensional systems, there are multiple feature functions, so we need to assume any linear combination of these feature functions does not vanish on a non-empty open set. We didn't present this assumption in our paper because we find it difficult to verify in practice. Nevertheless, we agree that this condition may contain a larger class than analytic functions, so we will include this and the corresponding discussions in the revised manuscript.
>
> &nbsp;
>
> ----
>
>
>
> We are happy to discuss more if the reviewer has remaining concerns or additional questions! Hope our response has addressed your concerns successfully! Look forward to hearing from you soon!

---

> > ### Comment · Reviewer_on2R · 2024-08-08
> >
> > I would like to thank the authors for their detailed response. I have decided to leave my score unchanged.

---

> > > ### Author Response · Authors · 2024-08-08
> > > **Thank you for your feedback!**
> > >
> > > Thank you very much for your feedback! Do you have remaining concerns or questions? We are more than happy to have further discussions!

---

### Official Review · Reviewer_Dfmt · 2024-07-17

**Soundness:** 4
**Presentation:** 4
**Contribution:** 3
**Rating:** 8
**Confidence:** 4

**Summary:**

The authors extend the work of Simchowitz et al. (2018) [linear] and Sattar et al. (2022) [bi-linear] to linear in the parameters but analytic nonlinear features showing that PE of inputs still results in PE of the states which for general nonlinear systems is not true. From this result LSE results like those originally in Simchowitz et al. (2018) are recovered for this new class of systems and SME results like those in Li et al. (2023a) are recovered for this broader class of systems.

**Strengths:**

- originality: original part is recognizing that analytics functions are only bad in finite number of spots and are otherwise "nice" functions
- quality: yes rigorous math
- clarity: well written
- significance: greatly extends the original results to a much larger class of systems

**Weaknesses:**

- the biggest weakness of the paper is that very little time is spent on the analysis in the main text. yes their is a proof sketch for the main theorem, but it would be nice to get more intuition from the main text.

**Questions:**

- I would much prefer the simulations in the appendix and the inclusion of the entirety of the proof of the main theorem in the main text with clear exposition of how the properties of analytic functions can be exploited to prove PE of u results in PE of x for these systems. This paper is not as bad as others where the main text is only an advertisement of the main result, but it would be a much tighter and more useful paper if this was done. This reviewer may be an outlier though as it seems the trend in these conferences is to do just the opposite of what I have suggested.

**Limitations:**

limitations are discussed

---

> ### Author Rebuttal · Authors · 2024-08-06
>
> Thank you very much for your constructive comments! We will address the comments below.
>
> ----
>
> `Weakness:`
> >The biggest weakness of the paper is that very little time is spent on the analysis in the main text. yes their is a proof sketch for the main theorem, but it would be nice to get more intuition from the main text.
>
> `Answer:` Thanks for your comment! We agree that the main text should provide more details and intuitions for the proofs of the main theorem. We will move the proof of Theorem 1 to the main text by taking advantage of the one additional page in the camera-ready version.
>
> -----
>
> `Question:`
> >I would much prefer the simulations in the appendix and the inclusion of the entirety of the proof of the main theorem in the main text with clear exposition of how the properties of analytic functions can be exploited to prove PE of u results in PE of x for these systems.
>
> `Answer:` Thanks for your suggestions! We will add more intuitions to the paper and move the simulations to the main text. Below is a brief discussion on intuitions. Firstly, notice that we only need to prove BMSB, which can be viewed as a stochastic version of PE. BMSB requires that any linear combination of feature functions should be positive with a non-vanishing probability. Since the linear combination of analytic functions is still an analytic function, and the zeros of an analytic function have measure zero, the probability that a linear combination of feature functions equal to 0 is also 0, as long as the noises follow semi-continuous distributions, which converts Lebesgue measure 0 to probability measure 0. We will incorporate this intuition together with more proof details in the revision.
>
> ----
>
> Thanks again for your helpful suggestions!

---

### Author Rebuttal · Authors · 2024-08-07

Thanks for your time and valuable comments! The attached is a pdf that contains new plots for addressing the questions from Reviewer wxwx and Reviewer QHWP.

Please feel free to ask us if you have any other questions. Looking forward to hearing your feedback!

---

### Comment · Area_Chair_rgqH · 2024-08-09
**Author-reviewer discussion**

Dear reviewers,
Thank you for your insightful feedback during the initial review. Kindly ensure that you review the author's rebuttal and, if necessary, engage in scientific conversation with them (the discussion period ends on Aug 13th). If the authors have satisfactorily addressed your concerns, please consider adjusting your scores. This is also an opportune moment to review feedback from other reviewers. Even if you find that your score remains unaltered, I would appreciate it if you could acknowledge reviewing the author's responses.
Once again, I truly appreciate your time and attention.
Best regards,
Area Chair

---

### Decision · Program_Chairs · 2024-09-25

**Decision:**

Accept (poster)

**Comment:**

This paper extends important results from linear and bi-linear system identification to a broader class of nonlinear systems by providing non-asymptotic guarantees for both least-squares estimators (LSE) and set-membership estimators (SME) within a framework that incorporates analytic functions. The work is rigorous, original, and well-presented, with reviewers particularly appreciating the clarity and significance of the theoretical contributions. While some concerns were raised regarding the restrictive nature of certain assumptions and the conservative nature of the theoretical bounds, the overall impact and novelty of the work still hold. Therefore, I recommend acceptance.